# JAK-STAT1 as therapeutic target for EGFR deficiency-associated inflammation and scarring alopecia

Karoline Strobl[1], Jörg Klufa[1], Regina Jin [ID][1], Lena Artner-Gent [ID][1], Dana Krauß [ID][1], Philipp Novoszel[1], Johanna Strobl[2], Georg Stary [ID][2], Igor Vujic [ID][3], Johannes Griss [ID][2], Martin Holcmann [ID][1], Matthias Farlik [ID][2], Bernhard Homey [ID][4], Maria Sibilia [ID][1✉] & Thomas Bauer [ID][1✉]

## Abstract

**The hair follicle stem cell niche is an immune-privileged micro-environment, characterized by reduced antigen presentation, thus shielding against permanent immune-mediated tissue damage. In this study, we demonstrated the protective role of hair follicle-specific epidermal growth factor receptor (EGFR) against scarring hair follicle destruction. Mechanistically, disruption of EGFR signaling generated a cell-intrinsic hypersensitivity within the JAK-STAT1 pathway, which, synergistically with interferon gamma expressing CD8 T-cell and NK-cell-mediated inflammation, compromised the stem cell niche. Hair follicle-specific genetic depletion of either JAK1/2 or STAT1 or therapeutic inhibition of JAK1/2 ameliorated the inflammation, restored skin barrier function and activated the residual stem cells to resume hair growth in mouse models of epidermal and hair follicle-specific EGFR deletion. Skin biopsies from EGFR inhibitor-treated and cicatricial alopecia patients revealed an active JAK-STAT1 signaling signature along with upregulation of antigen presentation and downregulation of key components of the EGFR pathway. Our findings offer molecular insights and highlight a mechanism-based therapeutic strategy for addressing chronic folliculitis associated with EGFR-inhibitor anti-cancer therapy and cicatricial alopecia.**

**Keywords** Cicatricial Alopecia; EGFR; Hair Follicle; Immune Privilege; JAK Inhibitors
**Subject Categories** Immunology; Skin; Stem Cells & Regenerative Medicine

See also: J S Durgin & S Y Wong

## Introduction

Hair serves an evolutionarily essential purpose in mammals by acting as a sensor, thermo-regulator and physical shield against external threats (Schneider et al, 2009). Hair follicles, however, also represent a vulnerable portal within the epidermal barrier. Mechanical or immunological dysfunction of this unit can be exploited by microbes leading to hair follicle inflammation and the consecutive destruction of this essential body structure (Polak-Witka et al, 2020).

To counter tissue destruction during inflammatory insults, hair follicles have evolved a relative immune privilege (IP) (Billingham and Silvers, 1971; Paus et al, 2003). It is characterized by low expression of major histocompatibility complex class I (MHC I) for self-tolerance, upregulation of "no danger" signals such as CD200, generation of an immunosuppressive microenvironment via TGF-β secretion, and recruitment of regulatory T cells, Trem2[+] macrophages, and invariant natural killer (NK) T cells (Agudo et al, 2018; Ali et al, 2017; Cohen et al, 2024; Liu et al, 2022; Wang et al, 2019; Wang et al, 2023).

Autoimmunity mediated by cytotoxic T cells can lead to the destruction of the hair follicle bulb region, which consists of transiently amplifying cells during hair growth, resulting in reversible hair loss known as alopecia areata (AA) in humans (Gilhar et al, 1998). Several anti-inflammatory drugs, including corticosteroids, calcineurin inhibitors, and janus kinase (JAK) inhibitors, have shown varying therapeutic effects in AA by targeting these autoreactive T-cells (Paus et al, 2018; Xing et al, 2014).

In contrast to AA, scarring (cicatricial) alopecia is characterized by the loss of the bulge region, which represents the hair follicle stem cell (HFSC) niche, leading to permanent and irreversible hair loss. Recent evidence implicates the collapse of the IP in HFSCs as a contributing factor in scarring alopecia types such as neutrophilic folliculitis decalvans and lymphocytic lichen planopilaris (Gilhar et al, 1998; Harries et al, 2018). Hallmarks of these conditions include initial stem cell hyper-proliferation, upregulation of MHC-I

[1]Center for Cancer Research, Medical University of Vienna and Comprehensive Cancer Center, Vienna 1090, Austria. [2]Department of Dermatology, Medical University of Vienna, Vienna 1090, Austria. [3]Department of Dermatology, Venereology and Allergy, Clinical Center Landstrasse, Vienna 1030, Austria. [4]Department of Dermatology, University Hospital Düsseldorf, Medical Faculty, Heinrich-Heine-University, Düsseldorf, Germany. ✉E-mail: maria.sibilia@meduniwien.ac.at; thomas.bauer@meduniwien.ac.at

and MHC-II, reduced CD200 expression, infiltration of various immune cells, subsequent apoptosis of HFSCs, hair shaft loss and finally the scarring of the hair follicle (Harries et al, 2020; Harries et al, 2013). So far, the therapeutic options for scarring alopecia are limited by the lack of mechanistic understanding.

In humans with a loss-of-function mutation in the epidermal growth factor receptor (EGFR) or ADAM17, an EGFR ligand sheddase, as well as patients undergoing long-term EGFR-inhibitor treatment during targeted cancer therapy, experience chronic hair and skin inflammation, which can be accompanied by scarring hair loss (Earl et al, 2020; Franzke et al, 2012; Nowaczyk et al, 2023; Satoh et al, 2020; Yang et al, 2011). The lack of effective treatments targeting the underlying causes of these severe cutaneous adverse events often leads to dose reduction or cessation of cancer therapy, compromising its efficacy (Holcmann and Sibilia, 2015; Lacouture, 2006).

Recently, we have demonstrated the importance of EGFR-ERK signaling in maintaining barrier integrity and preventing bacterial invasion and dysbiosis during hair shaft eruption, revealing the initial structural and inflammatory trigger (Amberg et al, 2019; Klufa et al, 2019; Lichtenberger et al, 2013; Mascia et al, 2013). We previously established that the barrier defect can be rescued by re-establishing active ERK signaling independent of EGFR through transgenic over-expression of SOS under the K5 promoter (EGFR$^{\Delta ep}$ K5-SOS). In addition, the microbial arm of the skin inflammation can be reduced by broad-spectrum antibiotic therapy (EGFR$^{\Delta ep}$ Abx). However, the molecular and immunological mechanisms driving and sustaining the chronic phase of the skin inflammation and its successive hair loss remain enigmatic.

Here we show that EGFR protects against microbiota-driven inflammatory destruction of the HFSC niche and subsequent scarring hair loss. Mechanistically, tissue destruction is initiated by the lack of EGFR-ERK signaling and driven by unleashed JAK-STAT1 activation in a cell-autonomous manner. CD8$^+$ T cells and NK cells expressing IFN-γ trigger the disruption of the HFSC niche. Prophylactic and therapeutic JAK1/2 inhibition disrupted the collapse of HFSC niche, restored hair growth, improved epidermal barrier function, and alleviated skin inflammation in epidermal and hair follicle-specific EGFR deleted mouse models. Active STAT1 signaling could be readily detected in patient samples during EGFR-inhibitor treatment and in different types of cicatricial alopecia.

These data represent mechanistic evidence of the hair follicle intrinsic JAK-STAT1 cascade being responsible for exhausting the stem cell niche during inflammation and implicates EGFR in securing its regulatory machinery. Our findings offer therapeutic strategies for effectively managing severe adverse events induced by EGFR inhibitors and scarring hair follicle destruction.

## Results

### Hair follicle-specific EGFR protects from microbiota-driven inflammatory depletion of the hair follicle stem cell niche

As previously published by our group, constitutive deletion of epidermal EGFR using K5-cre (EGFR$^{\Delta ep}$) results in epidermal barrier disruption and skin inflammation (Klufa et al, 2019). Interestingly, the reduction of the inflammation by transgenic K5-

SOS expression and antibiotic therapy (Abx) prevented visible hair loss between 2 and 5 months of age in EGFR$^{\Delta ep}$ mice (Fig. 1A). Additional FACS analysis revealed the loss of the CD34 positive hair follicle stem cell (HFSC) niche preceding macroscopic hair loss (Figs. 1A,B and EV1A for overall FACS gating strategy). The rescue of HFSCs during antibiosis is paralleled by a reduction of αβT cells (Fig EV1B). Most EGFR$^{\Delta ep}$ mice die within the first couple of weeks after birth due to the inflammation, which hinders the effective study of the late chronic inflammatory stage including hair loss (Klufa et al, 2019). Considering this, we generated a hair follicle-specific EGFR deletion mouse model using the Egr2-cre line (EGFR$^{\Delta Egr2}$, Fig. 1C–I) (Young et al, 2003). Efficient deletion of EGFR in the hair follicles and retained interfollicular epidermal EGFR expression was confirmed using immunofluorescence (IF) staining of EGFR in 3-month-old EGFR$^{\Delta Egr2}$ skin sections (Figs. 1C and EV1D). Most importantly, these mice develop early hair follicle-specific inflammation, as described by Langerhans cell (LC) specific MHC-II upregulation around the hair follicles at 2 weeks of age (p14) and later-occurring, microbiota-dependent hair loss, which is reversible upon Abx therapy, comparable to EGFR$^{\Delta ep}$ mice (Fig. 1D,E) (Bauer et al, 2012). The restriction of the inflammation to the hair follicles, however, leads to a dramatically improved survival of these mice over the pan-epidermal deletion model, which enabled us to study the events leading to the late-stage chronic skin inflammation and the subsequent hair loss in detail (Fig. EV1C). Chronological analysis revealed that the HFSC niche is established during morphogenesis and persists up to the first month after birth (Fig. 1F,H). However, starting at 2 months after the first anagen hair cycle phase, the HFSCs gradually disappeared over time with no detectable CD34 positive HFSCs remaining after 5 months concomitant with an epidermal influx of immune cells and visible hair loss (Fig. 1E,F). Moreover, we also observed a time-delayed size reduction of the entire hair follicle (Figs. 1G and EV1E–G, results from Figs. 1E–G and EV1G are summarized as illustration in Fig. 1H). Skin sections and epidermal whole mounts of older mice (5–7 M) indicated the complete degradation of the hair follicle structure together with the stem cell markers CD34, Sox9, and Krt15 (Figs. 1I and EV1F). These data demonstrate that hair follicle-specific EGFR protects from microbiota-driven inflammatory hair follicle stem cell attrition and alopecia.

### RNA profiling identifies hallmarks of scarring alopecia in EGFR-deficient CD34$^+$ hair follicle stem cells

Up to the first month of age, EGFR$^{\Delta Egr2}$ mice have a sufficient amount of HFSCs that can be readily detected, before starting to decrease (Figs. 1F and 2A). In order to analyze their transcriptional status at this initial time point, we isolated these cells using the CD34 HFSC surface marker from WT and EGFR$^{\Delta Egr2}$ mice at 1 month and performed RNA sequencing (RNAseq). Efficient *Egfr* deletion was verified by RT-PCR from the sorted cells and *Egfr* is the most significantly downregulated gene in the RNAseq dataset (Fig. EV2A,B). Principal component analysis revealed the dramatic transcriptional differences between WT and EGFR$^{\Delta Egr2}$ HFSCs (Fig. EV2C). Regulon analysis using DoRothEA revealed differential expression of transcription factors involved in cell cycle regulation (e.g., Foxm1, E2f2, E2f4 and Myc, Fig. 2B). In addition, among the top 50 deregulated genes (DEG) in EGFR$^{\Delta Egr2}$ HFSCs

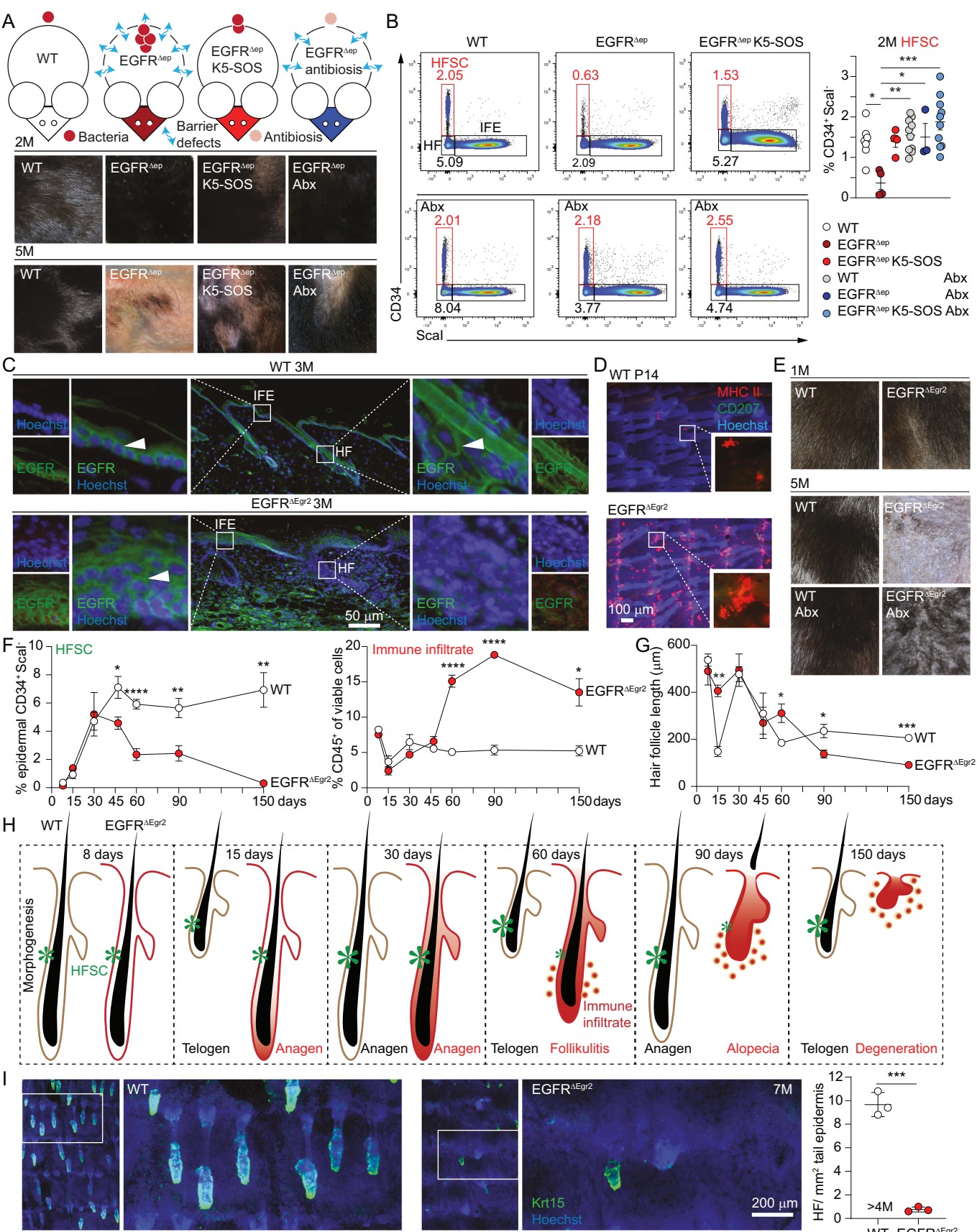

**Figure 1.   Hair follicle-specific EGFR protects from microbiota-driven inflammatory hair follicle stem cell loss.**

(A) Graphical summary of mouse models used and representative pictures of their back-skin at 2 months (2 M) and 5 M of age. Wildtype (WT) mice have an intact barrier and normal microbiota. EGFR$^{\Delta ep}$ mice develop a skin barrier defect and microbial dysbiosis. EGFR$^{\Delta ep}$ K5-SOS mice have an intact barrier but develop microbial dysbiosis (Klufa et al, 2019). Antibiotics (Abx, Cefazolin) treated EGFR$^{\Delta ep}$ mice are devoid of microbial inflammation while the initial barrier defect sustains. (B) Representative fluorescence-activated cell sorting (FACS) analysis of CD34$^+$ Sca-I$^-$ hair follicle stem cells (HFSC), hair follicles (HF) and interfollicular epidermis (IFE) of WT, EGFR$^{\Delta ep}$ and EGFR$^{\Delta ep}$ K5-SOS with or without antibiotic treatment at 2 M of age and the % of HFSCs quantified. Detailed gating strategy and inflammatory status see also Fig. EV1A,B. WT vs EGFR$^{\Delta ep}$ $p = 0.0201$, EGFR$^{\Delta ep}$ vs WT Abx $p = 0.0028$, EGFR$^{\Delta ep}$ vs EGFR$^{\Delta ep}$ Abx $p = 0.0434$, EGFR$^{\Delta ep}$ vs EGFR$^{\Delta ep}$ K5-SOS Abx $p = 0.0001$. (C) Immunofluorescence (IF) staining of skin sections of EGFR (green) in EGFR$^{\Delta Egr2}$ mice and their respective WT controls at 3 M. The white arrowheads mark examples for EGFR-positive cells. (D) LCs (CD207, green) and MHC-II$^{high}$ expression (red) in epidermal sheets from tails of WT and EGFR$^{\Delta Egr2}$ mice at P14. (E) Representative pictures of the hairy coat of EGFR$^{\Delta Egr2}$ at indicated time points and treatment. (F) Chronological change of the epidermal CD34$^+$ HFSCs and CD45$^+$ immune cells of EGFR$^{\Delta Egr2}$ and littermate WT controls as measured by FACS. HFSC day 45 $p = 0.0338$, HFSC day 60 $p = 0.0001$, HFSC day 90 $p = 0.0063$, HFSC day 150 $p = 0.0012$, CD45 day $p < 0.0001$, CD45 day 150 $p = 0.0226$. (G) Chronological analysis of the hair follicle length as measured from hematoxylin and eosin (H&E) stained skin sections of WT or EGFR$^{\Delta Egr2}$ mice. Data acquisition see also in EV1E. Day 15 $p = 0.0014$, day 60 $p = 0.0482$, day 90 $p = 0.0185$, day 150 $p = 0.0006$. (H) Illustration depicting timeline of hair cycle, immune infiltrate, loss of stem cells and degeneration of hair follicle. Based on data from (F, G) and Fig. EV1F,G. (I) Krt15 (green) expression and quantification in epidermal sheets from tails of WT and EGFR$^{\Delta Egr2}$ mice at 4–7 months. Data is presented in ±SEM, $p = 0.0001$, *$p < 0.05$, **$p < 0.01$, ***$p < 0.001$, ****$p < 0.0001$ by One-Way ANOVA with Tukey's posthoc correction, $n \geq 3$. Source data are available online for this figure.

was the downregulated *Bmp6*, implicated in HF quiescence and the upregulated proliferation-induced gene *Mki67* (Fig. EV2D). These data prompted us to specifically look at proliferation, HFSC activation and quiescence genes, which revealed the hyperproliferative status of the EGFR$^{\Delta Egr2}$ HFSCs compared to the anagen WT situation (Fig. 2C). Next, we confirmed these findings using IF to detect elevated numbers of Ki67 (i.e., proliferation maker) expressing CD34 positive HFSCs the EGFR$^{\Delta Egr2}$ mice at different time points to document the progression of the disease (Fig. 2D). In line with this, shaved dorsal skin of 2-month-old EGFR$^{\Delta Egr2}$ mice revealed that EGFR$^{\Delta Egr2}$ mice remained in the anagen hair cycle phase (dark color) at 2 months of age as opposed to the lighter telogen skin of WT mice (Fig. EV2E). Interestingly, gene set enrichment analysis (GSEA Wiki pathways), apart from proliferation, also identified signatures of apoptosis and fibrosis in these cells, already at 1 month (Fig. 2E). Therefore, we performed IF analysis from 3-month-old EGFR$^{\Delta Egr2}$ mice to investigate the progressive disease, which confirmed apoptotic (Caspase 8$^+$) HFSCs and Vimentin expression in the E-Cadherin positive hair follicle cells as a sign of fibrosis (Fig. 2F). After macroscopically visible hair loss at 5 months of age, skin sections of EGFR$^{\Delta Egr2}$ mice indicate follicular plugging and fibrotic hair structures, alpha smooth muscle actin expression and a dramatic decline of hair follicles after 10 months of age (Figs. 2G,H and EV2G,F). Taken together, we identified that during the chronic phase of skin inflammation, EGFR$^{\Delta Egr2}$ mice develop hallmarks of scarring hair follicle destruction (Harries et al, 2018).

## Hair follicle intrinsic inflammatory JAK-STAT1 signaling induces the antigen presentation machinery, hair follicle stem cell destruction and skin barrier disruption upon EGFR deletion

Next, we aimed to find the active signaling hubs in the HFSCs of EGFR$^{\Delta Egr2}$ mice. Cell signaling analysis from the HFSC RNAseq dataset with PROGENy identified the activity of the inflammatory JAK-STAT and TNFα pathways (Fig. 3A). Previous studies indicated that the TNFα pathway does not play a prominent role in the inflammatory EGFR$^{\Delta ep}$ phenotype (Lichtenberger et al, 2013; Mascia et al, 2010). In line with the PROGENy pathway analysis from our EGFR$^{\Delta Egr2}$ HFSC RNA profiling, we detected a broad induction of the JAK-STAT signaling hub, including downstream effectors (e.g., interferon induced

genes: *Ifi47*, *Ifi214*, and *Ifitm3*) and the downregulation of its negative regulator suppressor of cytokine signaling 3 (*Socs3*; Fig. 3B, first panel). SOCS3 protein reduction could also be visualized in the hair follicles from skin sections from EGFR$^{\Delta Egr2}$ mice as compared to the WT (Fig. EV2H). Notably, we also discovered a remarkable upregulation of various JAK-STAT utilizing receptor complexes on EGFR$^{\Delta Egr2}$ HFSCs (Fig. 3B second panel). This prompted us to further investigate possible downstream effectors. The JAK-STAT1 signaling cascade in keratinocytes (KCs) has been linked to MHC I and II upregulation (Limat et al, 1994; Shao et al, 2019). Indeed, we first identified the upregulation of genes encoding for the MHC I and II including its functional components (transporter associated with antigen processing: *Tap1* and *Tap2* genes) and regulators (Nod like receptor 5: *Nlrc5*) together with components of the immunoproteasome (e.g., proteasome subunit beta type-9, 8 and 10; Fig. 3B, third and fourth panel). MHC I and II protein upregulation were confirmed on the cell membrane of epidermal cells from EGFR$^{\Delta Egr2}$ mice in the chronic inflammatory phase as compared to the WT situation using FACS (Figs. 3C and EV2I for gating). Quiescent immune-privileged HFSCs have low MHC I expression and are able to resist pro-inflammatory signals to prevent its upregulation (Agudo et al, 2018). Especially MHC-I antigen presentation on HFSCs is considered as collapse of their immune privilege (Harries et al, 2018). EGFR$^{\Delta Egr2}$ HFSCs displayed MHC I and II upregulation on the RNA level and protein surface expression between 2 (postnatal day 15) and 4 (p30) weeks after birth (Figs. 3D and EV2J). In line with this, we could additionally detect the downregulation of the anti-inflammatory surface proteins CD55 and CD200 in EGFR$^{\Delta Egr2}$ mice (Fig. EV2K).

Next, we utilized our EGFR$^{\Delta ep}$ K5-SOS/antibiosis model system to investigate the role of barrier immunity versus microbiota in influencing MHC I and II regulation. The bacterial arm of skin inflammation predominantly induced HFSC-specific MHC-I surface expression, as only antibiotic treatment, not K5-SOS expression alone, preserved MHC-I expression (Fig. 3E). In order to interfere with the hair follicle specific JAK-STAT1 cascade and investigate its involvement in the MHC I and II expression status, skin function and inflammation, we crossed EGFR$^{\Delta Egr2}$ mice with either STAT1- or JAK1/2 floxed mice to generate double and triple hair follicle-specific knockout mice (Fig. 3F–H). Superficial hair loss was ameliorated in EGFR STAT1$^{\Delta Egr2}$ mice as compared to EGFR$^{\Delta Egr2}$ littermate controls (Fig. 3F). FACS analysis of the HFSCs revealed that their MHC-I expression is dependent on JAK-STAT1

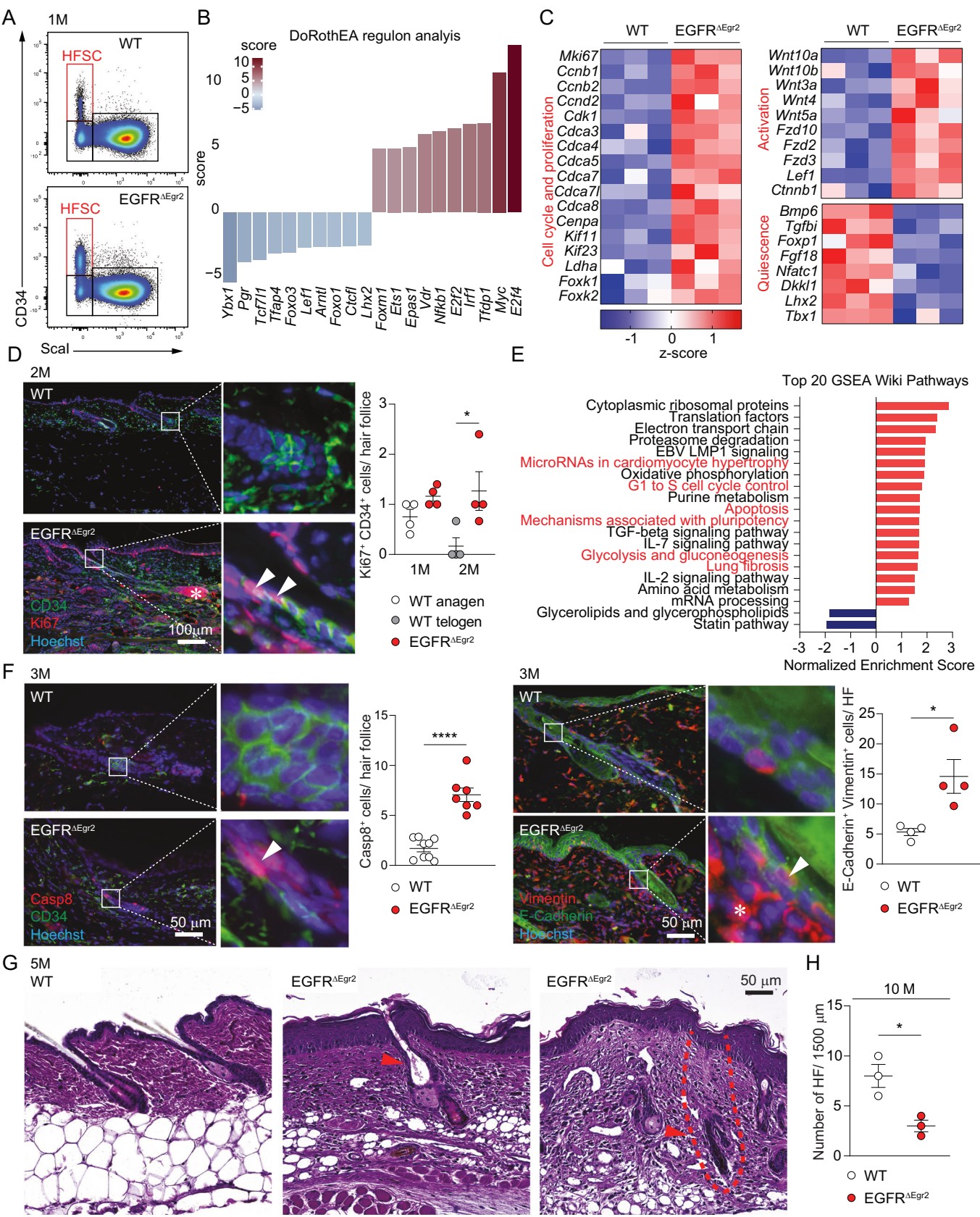

◄

**Figure 2. RNA profiling of EGFR-deficient hair follicle stem cells identifies hallmarks of scarring hair follicle destruction.**

(A) Representative FACS plot of sorting strategy for CD34$^+$Sca1$^-$ HFSCs (red box) of WT and EGFR$^{\Delta Egr2}$ mice at 1 M of age. (B) DoRothEA transcription factor analysis of the HFSC RNAseq dataset. (C) Heatmap with z-scores of selected differentially expressed genes of the HFSC RNAseq dataset from WT and EGFR$^{\Delta Egr2}$ mice. (D) IF analysis and quantification of proliferation (Ki67, red) in HFSCs (CD34, green) from skin sections of WT and EGFR$^{\Delta Egr2}$ mice at 1 M and 2 M of age ($n = 13$, 5 hair follicles per $n$). The white arrowheads mark examples for double-positive Ki67- and CD4-positive cells. Asterisk (*) highlights the additional proliferative bulb. WT vs EGFR$^{\Delta Egr2}$ 2 M $p = 0.0397$. (E) Top 20 gene set enrichment analysis (GSEA) WikiPathways from the HFSC RNAseq dataset. (F) Skin section IF staining of apoptosis (Caspase 8, red; $n = 16$, 3 hair follicles per $n$) and intermediate filament (Vimentin, red, E-Cadherin, green) expression in the hair follicle of WT and EGFR$^{\Delta Egr2}$ mice at 3 M of age ($n = 8$, 5 hair follicles per $n$). The white arrowheads mark double-positive Casp8- and CD34-positive or Vimentin- and E-Cadherin-positive cells. Asterisk (*) highlights Vimentin$^+$ dermal cells associated around the hair follicle. Casp8$^+$ WT vs EGFR$^{\Delta Egr2}$ $p < 0.0001$. Vim$^+$ WT vs EGFR$^{\Delta Egr2}$ $p = 0.018$. (G) H&E histochemistry of WT and EGFR$^{\Delta Egr2}$ at 5 M, red arrows indicate follicular plugging and a fibrotic hair follicle (indicated with a dotted line). WT vs EGFR$^{\Delta Egr2}$ $p = 0.0179$. (H) Number of hair follicle units per 1500 μm of H&E stained sections of WT and EGFR$^{\Delta Egr2}$ skin at 10 M of age. Data is presented in ±SEM, *$p < 0.05$, ****$p < 0.0001$ by unpaired t-test, $n \geq 3$. Source data are available online for this figure.

signaling (Fig. 3G, HFSC MHC-I). Most importantly, loss of HFSCs could be prevented by interrupting the cell-intrinsic JAK-STAT1 cascade (Fig. 3G, CD34 + HFSC). In addition, we also observed an overall amelioration of the skin inflammation as measured by epidermal CD45$^+$ immune cell infiltration and specifically αβ and γδT cell influx, which was paralleled by a reduction of the skin barrier defect (trans-epidermal water loss, TEWL) in the EGFR JAK1/2$^{\Delta Egr2}$ and EGFR STAT1$^{\Delta Egr2}$ mice, respectively (Fig. 3G,H). In summary, we conclude that the hair follicle-specific activation of the JAK-STAT1 cascade plays a dominant role in the collapse of the HFSC niche, hair follicle destruction and barrier immunity triggered by the lack of EGFR.

## Single-cell analysis revealed that immune recognition and HFSC destruction is triggered by IFNγ-expressing NK and CD8 T cells

The initial trigger of the inflammation is a structural barrier breach at the hair follicle accompanied by microbial invasion, with the latter inducing JAK-STAT mediated MHC I and II upregulation, prolonging the barrier defect and driving αβ and γδT cell recruitment (Fig. 3E–G) (Klufa et al, 2019). This indicates a microbiota-dependent immune effector, which triggers and feeds into the hair follicle JAK-STAT1 cascade. Therefore, we next sought to investigate the immune infiltrate and the epidermal cytokine/chemokine milieu. Chronically inflamed, bald EGFR$^{\Delta Egr2}$ mice at 5 months of age show a dramatic shift in their epidermal immune cell compartment as compared to the homeostatic dendritic epidermal T cell (DETC) and LC networks in WT mice as measured by FACS (Fig. 4A,B). They lose these steady-state compartments and acquire an immune infiltrate dominated by αβ-, γδT cells and neutrophils (Fig. 4A,B).

In order to expand the depth of this analysis, we performed single-cell RNA sequencing of the hair follicle and immune cell compartment of EGFR$^{\Delta Egr2}$ mice at 3 months of age, which confirmed the FACS data and additionally identified two types of innate lymphoid cells (ILCs), natural killer cells (NK cells) and allowed stratification of T-cells into CD4, CD8, and T regulatory cells (Tregs, Figs. 4C and EV3A). Confirming our data from Fig. 2, we also observed the fibrotic transition of the HFSC compartment (fibrotic bulge, FB) indicating scarring hair follicle destruction (Figs. 4C and EV3A). We next used this dataset to map the chemokine and cytokine profile of the immune- and hair follicle cell compartments (Fig. 4D). This captured the multifaceted inflammatory microenvironment in EGFR$^{\Delta Egr2}$ mice, with prominent *Il17*-producing γδT cells and ILC3s, *Il4* and *Il13* producing

CD4 Th2 cells and ILC2s, *Ifnγ* producing CD8 T-cells and NK cells and IL1α/β and OSM expressing macrophages. Given the importance of the JAK-STAT cascade in the hair follicles of EGFR$^{\Delta Egr2}$ mice, we next sorted out the possible JAK-STAT inducers IL4, IL13, IFN-γ, and OSM among the cytokine profile and screened for their potential in epidermal MHC I regulation using an ex vivo skin explant culture system (Figs. 4E and EV3B) (Bauer et al, 2021). We observed a significant upregulation of MHC I only with IFN-γ, which was further enhanced by the EGFR-inhibitor erlotinib (Fig. 4E). Putative cell-cell communication analysis using CellChat indicated that the only cellular sources of *Ifnγ* were the CD8$^+$ T and NK cells (Fig. 4F). CellChat also revealed a complex chemokine signaling network capable of recruiting the various T cell subsets and NK cells with *Ccl5* interlinking CD8$^+$ T cells and NK cells (Fig. 4F) (Homey et al, 2002). Interestingly, the gene signatures of the CD8 T cell and NK compartments cluster relatively close together (Figs. 4C and EV3C). Intracellular FACS analysis for IFN-γ confirmed its upregulation in infiltrating αβT cells and NK cells at the protein level (Fig. 4G).

In order to investigate the functional role of these cells during hair follicle destruction we used depletion antibodies for CD8 or NK1.1 in vivo. After confirming the cell deletion efficiency by the antibodies, we could show by FACS analysis, that both CD8 T cells and NK cell depletion led to a survival of the HFSCs concomitant with downregulated MHC I (Fig. EV3D–F and Fig. 4H). Interestingly, however, CD8 T cell and NK cell depletion did not impact on the γδT cell compartment and did not ameliorate the epidermal barrier defect as observed with JAK1/2 and STAT1 deficient hair follicles (compare Fig. 3G,H with Fig. 4I,J).

We, therefore, conclude that among the heterogeneous immune environment induced by prolonged EGFR deficiency, IFNγ-producing CD8 T cells and NK cells drive the hair follicle JAK-STAT1-dependent immune recognition and initiate the destruction of the HFSC niche. In parallel, EGFR deficiency leads to a hypersensitive JAK-STAT1 pathway in the hair follicle, which enhances its vulnerability during inflammation. Therefore, the pathogenesis consists of two separate insults acting synergistically.

## Therapeutic JAK inhibition re-initiates hair growth and ameliorates skin function and inflammation in EGFR$^{\Delta Egr2}$ and EGFR$^{\Delta ep}$ mice

We next tested the direct effect of IFN-γ on in vitro KCs, either from EGFR$^{\Delta ep}$ mice with their respective WT controls or WT mice treated with or without the EGFR-inhibitor erlotinib. MHC II induction was dramatically enhanced in EGFR-depleted or

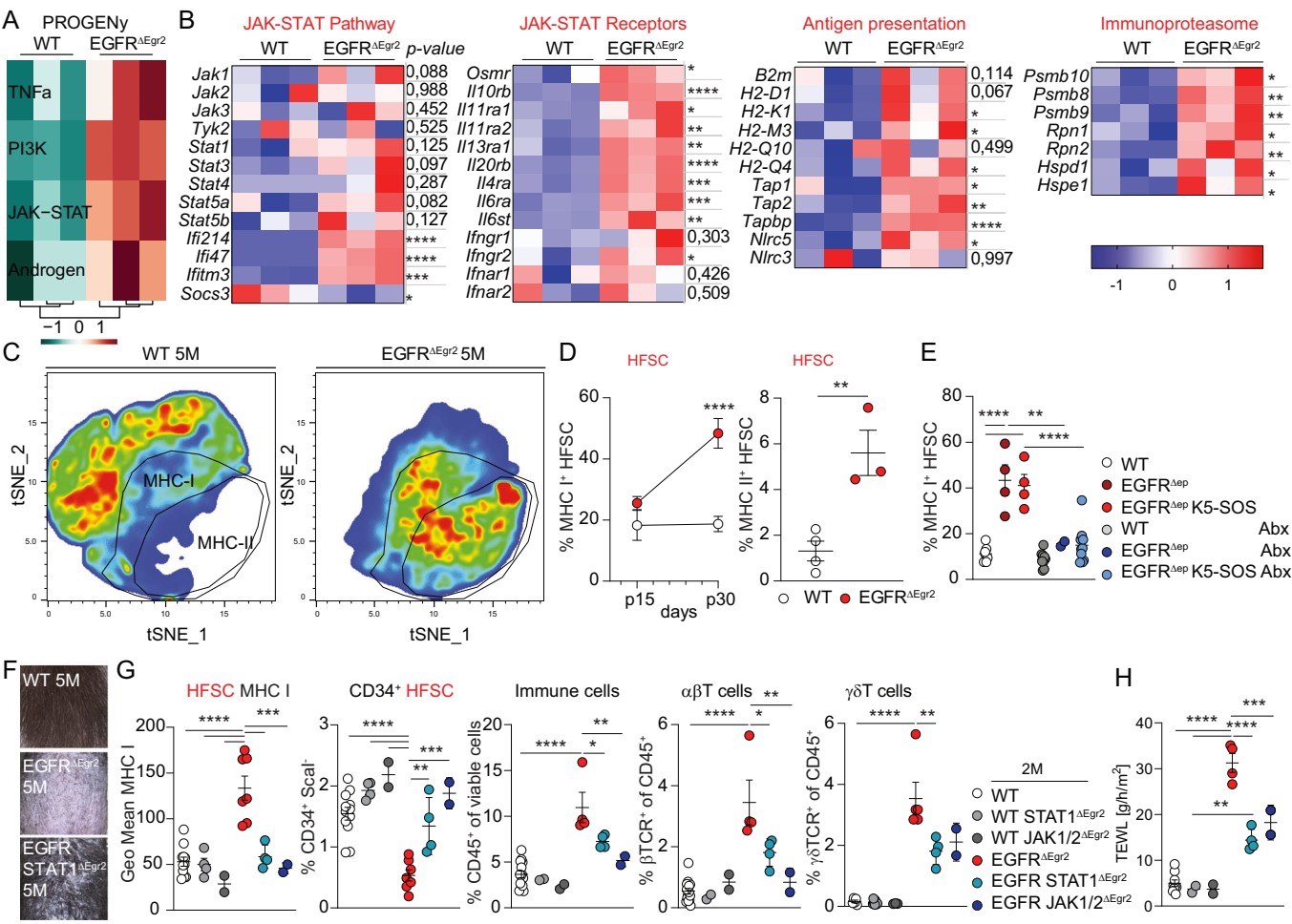

**Figure 3. Hair follicle intrinsic JAK-STAT1 signaling induces the antigen presentation machinery, drives hair follicle destruction and maintains barrier disruption upon EGFR deletion.**

(A) PROGENy analysis identifying upregulated signaling modules in the RNAseq dataset from sorted HFSCs from 1-month-old WT and EGFR$^{\Delta Egr2}$ mice. (B) Heatmap with z-scores and p-values between WT and EGFR$^{\Delta Egr2}$ mice of selected differentially expressed genes of the HFSC RNAseq dataset from WT and EGFR$^{\Delta Egr2}$ mice. *Ifi214* p = 0.0004, *Ifi47* p = 0.0005, *Ifitm3* p = 0.0006, *Socs3* p = 0.042, *Osmr* p = 0.0365, *Il10rb* p = 0.0001, *Il11ra1* p = 0.0148, *Il11ra2* p = 0.0108, *Il31ra1* p = 0.0025, *Il20rb* p = 0.0004, *Il4ra* p = 0.0008, *Il6ra* p = 0.0029, *Il6st* p = 0.0068, *Ifngr2* p = 0.0409, *H2-K1* p = 0.0161, *H2-M3* p = 0.0486, *H2-Q4* p = 0.0185, *Tap1* p = 0.0431, *Tap2* p = 0.007, *Tapbp* p = 0.0002, *Nlrc5* p = 0.0156, *Psmb10* p = 0.0251, *Psmb8* p = 0.0025, *Psmb9* p = 0.0099, *Rpn1* p = 0.0243, *Rpn2* p = 0.057, *Hspd1* p = 0.0479, *Hspe1* p = 0.0182. (C) tSNE FACS plot of MHC-I and MHC-II expression among CD45⁻ epidermal cells of WT or EGFR$^{\Delta Egr2}$ at 5 M. See also Fig. EV3 for gating. (D) Percentage of MHC-I or MHC-II expression on CD34⁺ HFSCs by flow cytometry in WT and EGFR$^{\Delta Egr2}$ at 1 M (p30 p = 0.0003 and WT vs EGFR$^{\Delta Egr2}$ p = 0.0069) and (E) in WT, EGFR$^{\Delta ep}$, EGFR$^{\Delta ep}$ K5-SOS and antibiotics (Abx) treated mice at 2 M. WT vs EGFR$^{\Delta ep}$ p < 0.0001, WT vs EGFR$^{\Delta ep}$ K5-SOS p < 0.0001, EGFR$^{\Delta ep}$ vs EGFR$^{\Delta ep}$ Abx p = 0.0012, EGFR$^{\Delta ep}$ K5-SOS vs EGFR$^{\Delta ep}$ K5-SOS Abx p < 0.0001. (F) Pictures of the hair phenotype of WT, EGFR$^{\Delta ep}$ and EGFR STAT1$^{\Delta Egr2}$ mice at the age of 5 M. (G) The geometric mean of MHC-I among CD34⁺Sca-I⁻ HFSCs (WT vs EGFR$^{\Delta Egr2}$ p < 0.0001, WT STAT1$^{\Delta Egr2}$ vs EGFR$^{\Delta Egr2}$ p < 0.0001, WT JAK1/2$^{\Delta Egr2}$ vs EGFR$^{\Delta Egr2}$ p < 0.0001, EGFR$^{\Delta Egr2}$ vs EGFR STAT1$^{\Delta Egr2}$ p = 0.0002, EGFR$^{\Delta Egr2}$ vs EGFR JAK1/2$^{\Delta Egr2}$ p = 0.0005), % of CD34⁺ HFSCs (WT vs EGFR$^{\Delta Egr2}$ p < 0.0001, WT STAT1$^{\Delta Egr2}$ vs EGFR$^{\Delta Egr2}$ p < 0.0001, WT JAK1/2$^{\Delta Egr2}$ vs EGFR$^{\Delta Egr2}$ p < 0.0001, EGFR$^{\Delta Egr2}$ vs EGFR STAT1$^{\Delta Egr2}$ p = 0.023, EGFR$^{\Delta Egr2}$ vs EGFR JAK1/2$^{\Delta Egr2}$ p = 0.0001, total CD45⁺ cells (WT vs EGFR$^{\Delta Egr2}$ p < 0.0001, EGFR$^{\Delta Egr2}$ vs EGFR STAT1$^{\Delta Egr2}$ p = 0.0454, EGFR$^{\Delta Egr2}$ vs EGFR JAK1/2$^{\Delta Egr2}$ p = 0.0063), αβT cells (WT vs EGFR$^{\Delta Egr2}$ p < 0.0001, EGFR$^{\Delta Egr2}$ vs EGFR STAT1$^{\Delta Egr2}$ p = 0.0171, EGFR$^{\Delta Egr2}$ vs EGFR JAK1/2$^{\Delta Egr2}$ p = 0.0014) and γδT cells (WT vs EGFR$^{\Delta Egr2}$ p < 0.0001, EGFR$^{\Delta Egr2}$ vs EGFR STAT1$^{\Delta Egr2}$ p = 0.0015) among CD45⁺ cells by flow cytometry of EGFR STAT1$^{\Delta Egr2}$ and EGFR JAK1/2$^{\Delta Egr2}$ at 2 M of age. (H) Transepidermal water loss (TEWL) measured on the back skin of WT, EGFR$^{\Delta Egr2}$, EGFR STAT1$^{\Delta Egr2}$ and EGFR JAK1/2$^{\Delta Egr2}$ mice at 2 M (WT vs EGFR$^{\Delta Egr2}$ p < 0.0001, WT STAT1$^{\Delta Egr2}$ vs EGFR$^{\Delta Egr2}$ p < 0.0001, WT vs EGFR STAT1$^{\Delta Egr2}$ p = 0.0002, WT STAT1$^{\Delta Egr2}$ vs EGFR STAT1$^{\Delta Egr2}$ p = 0.0027, EGFR$^{\Delta Egr2}$ vs EGFR STAT1$^{\Delta Egr2}$ p < 0.0001, EGFR$^{\Delta Egr2}$ vs EGFR JAK1/2$^{\Delta Egr2}$ p = 0.0004). Data is presented in ±SEM, *p < 0.05, **p < 0.01, ***p < 0.001, ****p < 0.0001 by unpaired t-test or One-Way ANOVA with Tukey's posthoc correction, n ≥ 2. Source data are available online for this figure.

inhibited KCs as compared to WT induction and this expression could be inhibited by the clinically approved JAK1/2 inhibitor ruxolitinib (Figs. 5A and EV4A). Encouraged by these results, we next applied the JAK1/2 inhibitor topically on 5-month-old bald EGFR$^{\Delta Egr2}$ mice to inhibit both, intrinsic JAK-STAT1 activation of the hair follicle and extrinsic IFN-γ and T cell activation in the immune compartment (Fig. 5B). Although, these mice only have a

rudimentary HFSC niche and already degraded hair follicles (see Fig. 2), we could, during the course of 28 days treatment, observe the induction of novel hair regrowth (Fig. 5B). This visible therapeutic effect was accompanied by the reduction of MHC surface expression and re-establishment of the HFSC niche (Fig. 5C,D). Interestingly, however, the re-activated HFSCs still had reduced expression levels of the mature stem cell surface

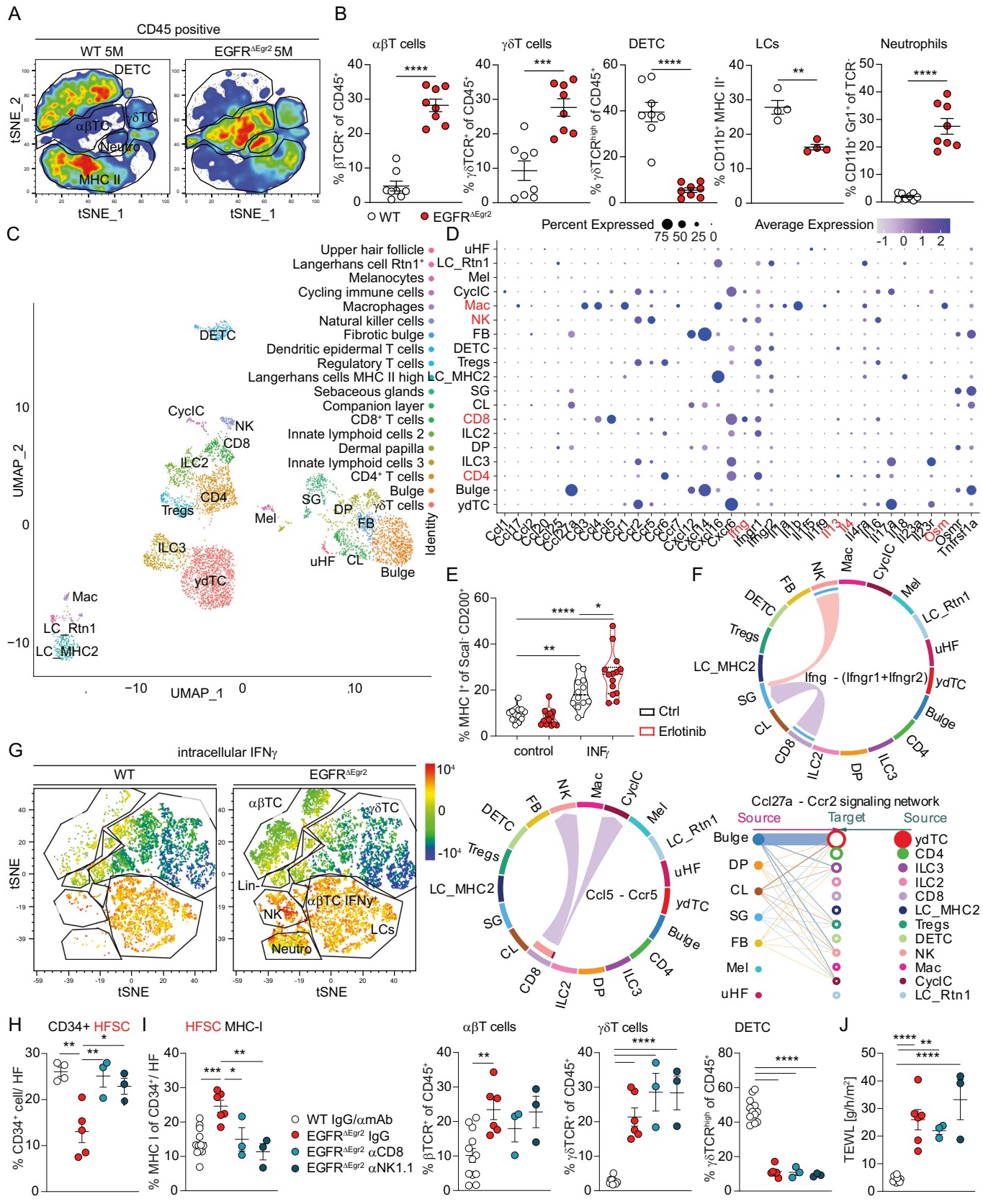

**Figure 4. Single-cell analysis revealed that immune recognition and HFSC destruction is triggered by IFNγ expressing NK and CD8 T cells.**

(A) tSNE FACS plot of epidermal CD45$^+$ immune cells of WT or EGFR$^{\Delta Egr2}$ at 5 M. See Fig. EV1A for gating strategy. (B) Quantification of indicated epidermal immune cells (αβT cells $p < 0.0001$, γδT cells $p = 0.0003$, DETC $p < 0.0001$, LCs $p = 0.0017$, Neutrophils $p < 0.0001$) by flow cytometry ($n = 16$). (C) UMAP of single-cell RNA sequencing analysis of epidermal CD45$^+$ immune cells and CD45$^-$ Sca-I$^-$ hair follicle cells of EGFR$^{\Delta Egr2}$ at 3 M of age, see also Fig. EV4 for population definition strategy. (D) Relative expression levels of selected cytokines and chemokines as bubble plot in cell subsets (possible JAK-STAT inducers highlighted in red). (E) Ex vivo WT skin explants treated with indicated cytokines with or without erlotinib for 48 h. FACS analysis of MHC-I expressions in CD200$^+$ HF cells. Control vs IFNγ $p = 0.0033$, ctrl vs erlotinib IFNγ $p < 0.0001$, INFγ vs erlotinib IFNγ $p = 0.0180$. (F) CellChat cell interaction analysis of the single-cell dataset. (G) tSNE FACS plot of the intracellular expression level of IFNγ expressing (positive in red) immune cells of WT or EGFR$^{\Delta Egr2}$ at 5 M of age. (H) Quantification of CD34$^+$ HFSCs of IF staining. WT IgG/αmAB vs EGFR$^{\Delta Egr2}$ IgG $p = 0.0025$, EGFR$^{\Delta Egr2}$ IgG vs EGFR$^{\Delta Egr2}$ αCD8 $p = 0.0079$, EGFR$^{\Delta Egr2}$ IgG vs EGFR$^{\Delta Egr2}$ αNK1.1 $p = 0.0278$. (I) Percentage of MHC-I expression on CD34$^+$Sca-I$^-$ HFSCs (WT IgG/αmAB vs EGFR$^{\Delta Egr2}$ IgG $p = 0.0004$, EGFR$^{\Delta Egr2}$ IgG vs EGFR$^{\Delta Egr2}$ αCD8 $p = 0.0217$, EGFR$^{\Delta Egr2}$ IgG vs EGFR$^{\Delta Egr2}$ αNK1.1 $p = 0.0015$) and % of αβT cells (WT IgG/αmAB vs EGFR$^{\Delta Egr2}$ IgG $p = 0.0066$), γδT cells (all $p < 0.0001$) and dendritic epidermal T cells (DETC, all $p < 0.0001$) among CD45$^+$ cells by flow cytometry from WT and EGFR$^{\Delta Egr2}$ depleted of CD8 T cells or NK1.1 cells and their respective controls. (J) Transepidermal water loss (TEWL) in CD8 T cell or NK1.1 cell depleted WT and EGFR$^{\Delta Egr2}$ mice (WT IgG/αmAB vs EGFR$^{\Delta Egr2}$ IgG $p < 0.0001$, WT IgG/αmAB vs EGFR$^{\Delta Egr2}$ αCD8 $p = 0.0019$, WT IgG/αmAB vs EGFR$^{\Delta Egr2}$ αNK1.1 $p < 0.0001$). Data is presented in ±SEM, *$p < 0.05$, **$p < 0.01$, ***$p < 0.001$, ****$p < 0.0001$ by unpaired t-test or One-Way ANOVA with Tukey's posthoc correction, $n \geq 3$.

marker CD34 but could be readily detected by the expression of SOX9 (Figs. 5D and EV4B) (Watt and Jensen, 2009). Prophylactic ruxolitinib treatment starting from 1-month-old EGFR$^{\Delta Egr2}$ mice, however, prevented the loss of CD34 surface expression, similar to the JAK1/2 or STAT1 deletion in this model (Fig. EV4C). In the therapeutic setting in 5-month-old mice, the re-appearance of SOX9-positive HFSCs occurs concomitantly with an increase in hair follicle length (Fig. 5E). Apart from novel hair regrowth, therapeutic JAK1/2 inhibition was able to restore the epidermal barrier function, normalize epidermal thickness and stopped αβT cell infiltration (Fig. 5F–H). Interestingly, JAK1/2 inhibition did not influence the γδT cell compartment, the neutrophil recruitment and the NK cells (Fig. EV4D).

In order to extrapolate the mechanism and the therapeutic protocol to the full-scale skin inflammation model, we applied the JAK1/2 inhibitor to the right side (including the ear) of mice lacking EGFR in the complete epidermal compartment (EGFR$^{\Delta ep}$ mice) with the left side treated with vehicle control (Fig. 5I,J). Similar to the results from the EGFR$^{\Delta Egr2}$ folliculitis model, topical JAK1/2 inhibition re-established skin barrier function, restored visible hair growth and reduced MHC expression on the treated side (right side) of the mice (Fig. 5I,J). Especially the restored epidermal barrier function and the hair regrowth indicate the successful preclinical therapeutic approach to treat the full-blown skin inflammation. We next crossed EGFR$^{\Delta ep}$ mice with STAT1 floxed animals and could observe similar prophylactic effects as with the EGFR$^{\Delta Egr2}$ hair follicle model (compare Fig. 3F–H to Fig. 5K–M). In contrast to the prophylactic and the therapeutic setting (treatment initiated in 5-month-old mice), fully scarred 10-month-old EGFR$^{\Delta ep}$ mice did not exhibit hair regrowth in response to JAK inhibitor treatment (Fig. EV4E,F).

We could previously demonstrate that EGFR-controlled ERK signaling prevents the initial barrier disruption during novel hair shaft eruption or outgrowth (Klufa et al, 2019). In order to investigate the involvement of the JAK cascade during this initial structural insult, we treated EGFR$^{\Delta ep}$ mice during hair eruption (starting from p6 until p19) topically with the JAK1/2 inhibitor (Fig. EV4G,H). As expected, this early prophylactic JAK1/2 inhibition did not impact on the initial ERK-dependent barrier breach, but only ameliorated MHC II expression and prevented αβT cell influx. This indicates that the early barrier disruption is independent of JAK1/2 signaling and that hyper-activated JAK-STAT1 drives the chronic phase of hair and skin inflammation.

Taken together, we could demonstrate the effectiveness of topical JAK inhibition in reducing skin inflammation, restoring epidermal barrier function and prevention of hair follicle destruction caused by prolonged EGFR dysfunction. These data represent preclinical evidence for the therapeutic potential of JAK inhibitors to manage adverse events during EGFR-inhibitor-targeted cancer therapy and reverse the development of scarring hair follicle destruction.

## Active STAT1 signaling in patients with scarring alopecia

In order to confirm the most important hallmarks of our findings in patients, we next analyzed biopsies of EGFR-inhibitor-treated squamous cell carcinoma (SCC) cancer patients. Comparison of pre- and post-EGFR-inhibitor treatment skin biopsies confirmed epidermal MHC I upregulation (Fig. 6A). Furthermore, we detected elevated phosphorylated STAT1 protein in the clinical samples after EGFR inhibition, indicating its chronic activation (Fig. 6B).

To extrapolate our findings to scarring alopecia, we screened the here identified key features of our mouse model in a spatial transcriptomic dataset from two patients with lichen planopilaris (LPP) and healthy scalp skin (Figs. 6C and EV5A) (Data Ref: Cohen et al, 2024). We observed the upregulation of the fibrotic markers COL1A1 and FN1 and the downregulation of stem cell marker KRT15 around the hair follicles, identifiable by KRTDAP expression and H&E staining, in the LPP biopsies (Figs. 6C and EV5A). Notably, EGFR and its ligand AREG are markedly downregulated in these patients. In line with our mouse model, we noted the upregulation of MHC-I (B2M), antigen processing TAPBP and JAK-STAT signature genes STAT1 and JAK1 (Figs. 6C and EV5A).

Confirming these transcriptomic data, phosphorylated STAT1 was elevated on protein level in the hair follicle and epidermis of biopsies from folliculitis decalvans (FD), frontal fibrosing alopecia (FFA) and LPP patients (Figs. 6D,E and EV5B,C).

These results emphasize the significance of targeting the JAK-STAT1 signaling hub as a crucial therapeutic strategy in the context of EGFR inhibition and the treatment of human scarring alopecia.

## Discussion

The hairy coat is a survival necessity of mammals, as it protects from life-threatening outer influences like cold or ultraviolet

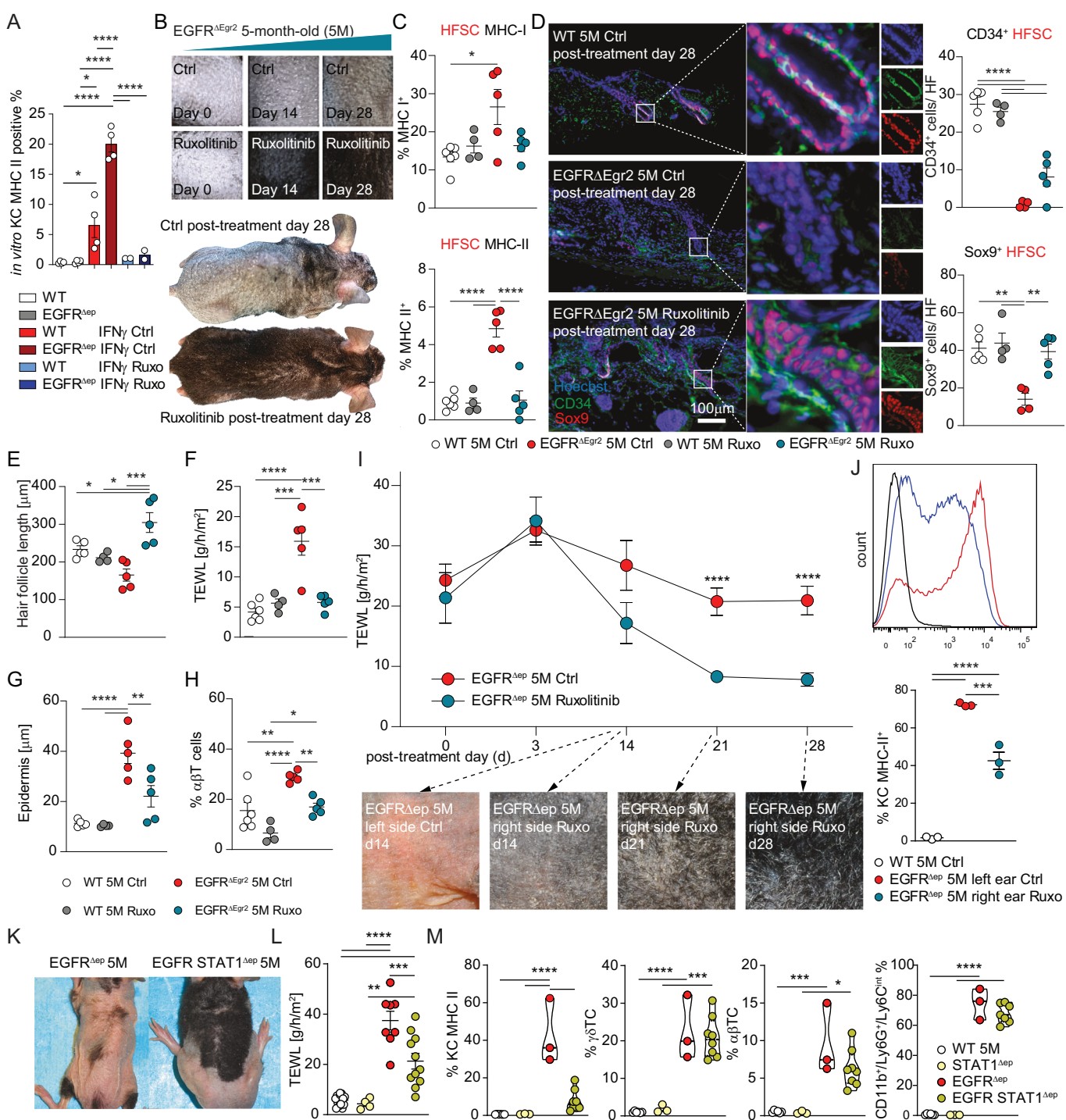

radiation. In humans, reversible or irreversible hair loss, irrespective of the cause, severely affects the Quality-of-Life and psychological well-being of patients (Chiang et al, 2015).

Using hair follicle-specific JAK1/2 and STAT1 knockout mice, we expanded the spectrum of activity of the JAK inhibitors to HFSC-specific protection from chronic inflammation resulting in scarring alopecia. As this condition is considered irreversible and current treatment regimens are only able to delay hair loss, it is of clinical importance that we were able to prophylactically prevent

but also therapeutically reverse developing scarring hair follicle destruction using JAK inhibitors in mice.

This suggests that not all HFSCs are destroyed simultaneously, but rather that the HFSC niche is constantly degraded and scarred over time. We observed a decline in HFSCs at 2 month of age during the prolonged anagen phase of the EGFR^ΔEgr2 and the start of visible hair loss from 3-month-old EGFR^ΔEgr2 mice onward. Despite that, we were able to effectively regain hair growth up to 5 months of age with therapeutic JAK inhibition. In older mice (e.g.,

**Figure 5. Therapeutic JAK inhibition re-initiates hair growth and ameliorates skin function and inflammation in EGFR$^{\Delta Egr2}$ and EGFR$^{\Delta ep}$ mice.**

(A) MHC-II expression as analyzed by flow cytometry of in vitro EGFR$^{\Delta ep}$ or WT primary murine keratinocytes treated with and without IFNγ and JAK1/2 inhibitor (ruxolitinib) as indicated (WT ctrl vs WT IFNγ $p = 0.0189$, EGFR$^{\Delta ep}$ vs WT IFNγ $p = 0.021$, all other indicated comparisons $p < 0.0001$). (B) Representative pictures of hair growth initiation in EGFR$^{\Delta Egr2}$ mice treated topically with DMSO (vehicle ctrl) or 3% ruxolitinib in DMSO daily at the age of 5 M for 4 weeks. (C) Percentage of MHC-I or MHC-II expression in HFSCs by flow cytometry (% MHC I $p = 0.0101$, % MHC II all values $p < 0.0001$), (D) Immunofluorescence staining and quantification of CD34$^+$ ($p < 0.0001$) and Sox9$^+$ (WT vs EGFR$^{\Delta Egr2}$ $p = 0.0016$, WT Ruxo vs EGFR$^{\Delta Egr2}$ $p = 0.0011$, EGFR$^{\Delta Egr2}$ vs EGFR$^{\Delta Egr2}$ Ruxo $p = 0.0028$) HFSCs, ($n = 18$, 3 hair follicles per $n$) (E) Hair follicle length as measured from H&E stained skin sections (WT vs EGFR$^{\Delta Egr2}$ Ruxo $p = 0.0417$, WT Ruxo vs EGFR$^{\Delta Egr2}$ Ruxo $p = 0.0108$, EGFR$^{\Delta Egr2}$ vs EGFR$^{\Delta Egr2}$ Ruxo $p = 0.0002$), (F) TEWL measurement (WT vs EGFR$^{\Delta Egr2}$ Ruxo $p < 0.0001$, WT Ruxo vs EGFR$^{\Delta Egr2}$ Ruxo $p = 0.0004$, EGFR$^{\Delta Egr2}$ vs EGFR$^{\Delta Egr2}$ Ruxo $p = 0.0002$), (G) Epidermal thickness measured from H&E stained skin sections (WT vs EGFR$^{\Delta Egr2}$ Ruxo and WT Ruxo vs EGFR$^{\Delta Egr2}$ $p < 0.0001$, EGFR$^{\Delta Egr2}$ vs EGFR$^{\Delta Egr2}$ Ruxo $p = 0.0063$) and (H) % of αβT cells (WT vs EGFR$^{\Delta Egr2}$ $p = 0.0027$, WT Ruxo vs EGFR$^{\Delta Egr2}$ $p < 0.0001$, EGFR$^{\Delta Egr2}$ vs EGFR$^{\Delta Egr2}$ Ruxo $p = 0.0096$, WT Ruxo vs EGFR$^{\Delta Egr2}$ Ruxo $p = 0.0366$) among CD45$^+$ immune cells by flow cytometry from EGFR$^{\Delta Egr2}$ mice treated topically with DMSO (vehicle ctrl) or 3% ruxolitinib in DMSO daily at the age of 5 M for 4 weeks and the respective WT controls. (I) TEWL and representative pictures of EGFR$^{\Delta ep}$ mice treated with 3% ruxolitinib on the right side and DMSO (vehicle ctrl) on the left side over the 4-week treatment period (day 21 and 28 $p < 0.0001$). (J) Percentage of MHC-II expression in CD45 negative keratinocytes by flow cytometry of EGFR$^{\Delta ep}$ treated with DMSO (vehicle ctrl) or 3% ruxolitinib on the left and the right ear, respectively (WT vs EGFR$^{\Delta ep}$ and WT vs EGFR$^{\Delta Egr2}$ Ruxo $p < 0.0001$, EGFR$^{\Delta ep}$ vs EGFR$^{\Delta ep}$ Ruxo $p = 0.0006$). (K–M) WT, EGFR$^{\Delta ep}$ and EGFR STAT1$^{\Delta ep}$ mice were analyzed for their hairy coat (K), TEWL (WT vs EGFR$^{\Delta ep}$ STAT1, WT vs EGFR$^{\Delta ep}$ and WT STAT1 vs EGFR$^{\Delta ep}$ $p < 0.0001$, WT STAT1 vs EGFR STAT1$^{\Delta ep}$ $p = 0.0028$, EGFR$^{\Delta ep}$ vs EGFR STAT1$^{\Delta ep}$ $p = 0.0003$) (L) and for their inflammatory status by FACS for the indicated parameters (KC MHC II $p < 0.0001$, γδTC WT vs EGFR$^{\Delta ep}$ $p < 0.0001$ and WT STAT1 vs EGFR STAT1$^{\Delta ep}$ $p = 0.0001$, αβTC WT vs EGFR$^{\Delta ep}$ $p = 0.0003$ and WT STAT1 vs EGFR STAT1$^{\Delta ep}$ $p = 0.0101$, CD11b/Ly6G$^+$/Ly6C$^{int}$ % $p < 0.0001$) (M). Data is presented in ±SEM, *$p < 0.05$, **$p < 0.01$, ***$p < 0.001$, ****$p < 0.0001$ by One-Way ANOVA with Tukey's posthoc correction, $n \geq 2$. Source data are available online for this figure.

10 months of age), the complete scarring of the hair follicle then prevents therapeutic intervention indicating the irreversible loss of the stem cell niche. This represents evidence of a therapeutic window for treating scarring hair loss, although, in the various human conditions, it may differ greatly depending on the intensity of the inflammatory insult and the source and potency of the respective JAK-STAT1 activator.

It has been described that JAK inhibitors, although effective in some autoimmune diseases like rheumatoid arthritis or AA, do not necessarily have a clinically stable effect and after the cessation of the therapy, relapses are very common (Horesh et al, 2024; Kennedy Crispin et al, 2016; Tanaka et al, 2022). This might reflect the only temporary inhibiting effect on autoreactive T cells. However, in scarring alopecia a more stable or longer-lasting therapeutic effect could be possible in part due to the here-described multidirectional mode of action, which includes cell-intrinsic restoration of the stem cell niche, blunting the recruitment and activation of the immune effectors, and by the restoration of skin barrier function. In addition, other case studies report successful off-label JAK inhibitor therapy in cases of some scarring alopecia types (Chen et al, 2024; Jerjen et al, 2020; Moussa et al, 2022; Yang et al, 2018). We now present the mechanistic explanation for these unbiased treatment attempts. Clinical trials are underway to disclose, whether JAK inhibitors can impact scarring alopecia in the long-term (NCT05076006, NCT05549934).

Upon EGFR inhibition, the initial immune response does not require the adaptive arm of the immune system. However, during the chronic phase, microbial dysbiosis leads to the recruitment of various T cell subsets to the epidermal compartment. Their cytokine profile indicates Th1, 17 and 2 responses, which are supported by their corresponding innate lymphoid cells, similar to what can be observed in atopic dermatitis (AD) (Howell et al, 2023; Vasquez Ayala et al, 2023). Depleting CD8$^+$ T cells expressing IFN-γ identified the driver of hair follicle-specific JAK-STAT1 activation and re-established the HFSC IP. In addition, we identified NK cells as IFN-γ producers, which were synergistically disrupting the IP in our EGFR-specific model system.

Interestingly, transcriptional profiling of the HFSCs revealed a broad upregulation of JAK-STAT potent receptors. Among them

the IL4- and IL13 receptors. Recent clinical case studies indicate that blocking IL4 and IL13 during AD-associated alopecia can reverse the hair loss indicating that various JAK-STAT inducers can feed into the HFSC intrinsic cascade (Guttman-Yassky et al, 2022; Howell et al, 2023). This additionally indicates that HFSC-specific inhibition of the JAK-STAT1 cascade might act universally and is independent of the driving cytokines and its cellular source.

Therefore, it would be beneficial for patients with scarring alopecia, irrespective of the type, to screen for therapeutic markers highlighting JAK-STAT1 activity or JAK-inhibitor sensitivity to predict treatment efficiency or success. Common genetic signatures of cicatricial alopecia patients, identified fibrosis, immune cell pathways and susceptibility of frontal fibrosing alopecia with a locus containing the MHC-I region (Tziotzios et al, 2019; Wang et al, 2022). We could readily visualize the phosphorylation of STAT1 and the upregulation of MHC-I on epidermal cells from patients under EGFR-inhibitor therapy and the pSTAT1 signature in patients with cicatricial alopecia. These might represent possible candidates and readily available diagnostic markers for pre-screening patients with hair loss. However, more stable targets downstream of JAK-STAT1 like the interferon induced proteins like IFITM3 or soluble indicators measurable in liquid biopsies could help to effectively stratify patients in the future.

The molecular mechanism by which EGFR regulates the JAK-STAT1 sensitivity of epidermal cells might be ultimately driven by cell-intrinsic JAK-STAT receptor over-expression and simultaneous down-modulation of negative feedback mechanisms like SOCS3 as we could show here. EGFR might play a role in the overall downregulation of interferon pathways, since virus-activated EGFR has also been shown to suppress IFN signaling during viral infection, thereby decreasing antiviral defense (Ueki et al, 2013). EGFR, apart from regulating epidermal ERK and AKT signaling, is known to feed into the STAT3 cascade. Therefore, it is possible that the decline of epidermal EGFR-mediated STAT3 signaling switches the balance in favor of STAT1 activation as it is described for STAT3 knockout conditions in various model systems including cancer cells (Concha-Benavente et al, 2013).

Furthermore, we demonstrate a hyper-proliferative state in EGFR-deficient hair follicle stem cells leading to a prolonged

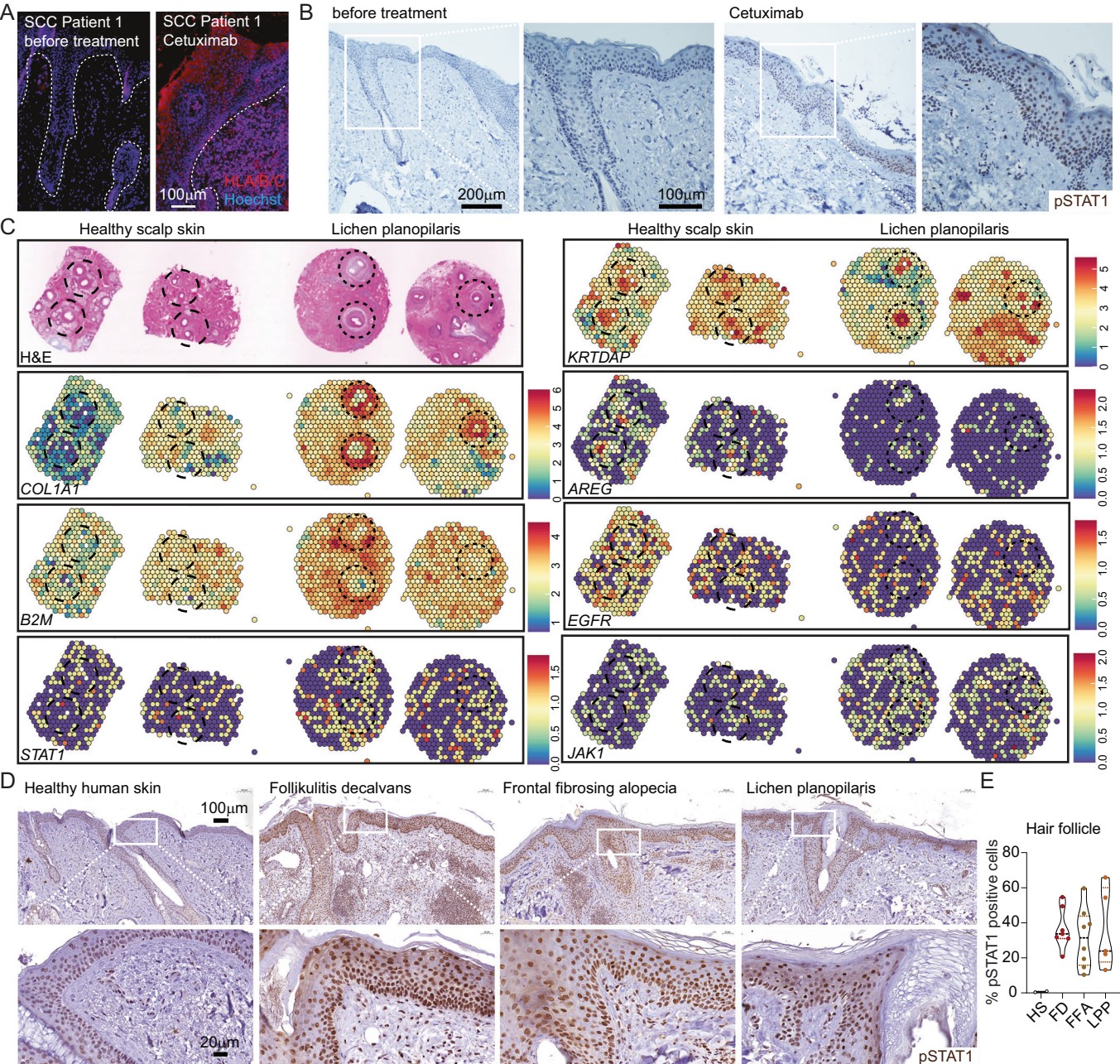

**Figure 6. Active JAK-STAT1 signaling signature in patients with scarring alopecia.**

(A) Human skin sections from biopsies of patients before EGFR-inhibitor treatment (left side) and after (right side) stained for MHC-I (red) and Hoechst (blue). Dotted line defines epidermal–dermal border. Images are representative pictures from 2 independent patients. (B) Human skin sections stained immunohistochemically for phosphorylated STAT1 protein (pSTAT1) before EGFR-inhibitor treatment and after. Images are representative pictures from 2 independent patients. (C) Spatial transcriptomic feature plots of the indicated genes superimposed on the corresponding H&E images. Dataset from Cohen et al (Data ref: Cohen et al, 2024). (D) Human skin sections from biopsies of patients before EGFR-inhibitor treatment (normal human skin) and patients with folliculitis decalvans (FD), frontal fibrosing alopecia (FFA) and lichen planopilaris (LPP, as indicated on top) stained for pSTAT1 immunohistochemically. Scale bar is 100 μm for top panel and 20 μm for lower panel. (E) Quantification of pSTAT1 staining intensity as analyzed from (D). Each dot represents an individual patient sample. See also in EV5B and C. Source data are available online for this figure.

anagen phase, potentially to counteract apoptosis, inflammation, wound healing, and fibrosis. As only quiescent stem cells evade immune surveillance, this compensatory mechanism could contribute to the overall loss of IP (Agudo et al, 2018). It is intriguing to speculate that similar mechanisms of IP preservation in HFSCs, identified in this study, are also active in various EGFR-dependent solid tumors and that EGFR inhibition renders them more immunogenic, thereby supporting anti-cancer therapy efficiency. Indeed, it is a long-standing observation that the overall survival of cancer patients positively correlates to their

EGFR-targeted anti-cancer therapy's adverse events (Pérez-Soler, 2003). Our study, however, indicates that the cell intrinsic JAK-STAT1 sensitivity introduced by the lack of EGFR also needs to be triggered and driven by the immune cell compartment, which opens room for improving or boosting the therapy effect by JAK activators.

Vice versa, further studies need to evaluate the feasibility of using JAK-inhibitors to treat EGFR-inhibitor induced adverse events, as JAK-inhibitors are predicted to rather support cancer growth by blunting the activity of cytotoxic T-cells. However, an anti-tumorigenic role has been attributed to STAT3, not STAT1, signaling in cytotoxic CD8 T cells (Sun et al, 2023). Therefore, topical application of specific JAK-inhibitors in this setting is crucial for the benefit of the patient and cancer therapy outcome.

Taken together, our study identifies EGFR as cell intrinsic immune modulator, which tightens the strings on the JAK-STAT1 cascade to prevent inflammatory destruction of evolutionary important tissues like the hair follicle.

# Methods

### Reagents and tools table

| Reagent/Resource | Reference or Source | Identifier or Catalog Number |
| --- | --- | --- |
| **Experimental Models** | | |
| Mouse: K5cre x EGFRfl/fl | Lichtenberger et al, 2013 | N/A |
| Mouse: K5-SOS | Sibilia et al, 2000 | N/A |
| Mouse: K5cre x EGFRfl/fl, K5-SOS transgenic | Klufa et al, 2019 | N/A |
| Mouse: Egr2cre (Egr2tm2(cre)Pch/J) | The Jackson Laboratory | Strain #:025744 |
| Mouse: Egr2cre x EGFRfl/fl | This study | N/A |
| Mouse: JAK1fl/fl (C57BL/6N-Jak1tm1c(EUCOMM)Hmgu/H) | Dr. Alexander Dohnal | N/A |
| Mouse: JAK2fl/fl | Wagner et al, 2004; Krempler et al, 2004 | N/A |
| Mouse: STAT1fl/fl | Wallner et al, 2012 | N/A |
| Mouse: Egr2cre x JAK1fl/fl JAK2fl/fl EGFRfl/fl | This study | N/A |
| Mouse: Egr2cre x STAT1fl/fl EGFRfl/fl | This study | N/A |
| **Antibodies** | | |
| Armenian hamster anti-mouse TCR γ/δ Antibody, FITC, 1:200 | Biolegend | Cat# 118106, RRID: AB_313830 |
| Rat anti-mouse/human CD11b Antibody, Pacific Blue, 1:200 | Biolegend | Cat# 101224, RRID: AB_755986 |
| Rat anti-mouse Ly-6G Antibody anti-Ly-6G, Alexa Fluor 700, 1:100 | Biolegend | Cat# 127622, RRID: AB_10643269 |
| Rat anti-mouse I-A/I-E Antibody, APC/Cy7, 1:400 | Biolegend | Cat# 107628, RRID: AB_2069377 |
| Armenian hamster anti-mouse TCR β chain Antibody, PE, 1:200 | Biolegend | Cat# 109208, RRID: AB_313431 |
| Rat Anti-Langerin/CD207 Antibody (929F3.01), Alexa Fluor 488, 1:200 | Dendritics | Cat# DDX0362A488, RRID: AB_1148740 |
| Rat anti-mouse I-A/I-E antibody, PE, 1:400 | Biolegend | Cat# 107608, RRID: AB_313323 |
| Rat anti-mouse/human langerin/CD207, Alexa Fluor 488, 1:200 | Dendritics | Cat# DDX0362 929F3.01 |

| Reagent/Resource | Reference or Source | Identifier or Catalog Number |
| --- | --- | --- |
| Rabbit anti-mouse Ki67, 1:200 | Abcam | Cat# ab15580, RRID: AB_443209 |
| Rabbit anti-mouse Vimentin, 1:100 | Cell Signalling | Cat# 5741, RRID: AB_10695459 |
| Rabbit anti-mouse Caspase 8, 1:200 | Abcam | Cat# ab25901, RRID: AB_448890 |
| Rabbit anti-mouse EGFR, 1:1000 | Bio-Techne | Cat# AF1280, RRID: AB_354717 |
| Rat anti-mouse E-Cadherin, 1:200 | Abcam | Cat# ab11512, RRID: AB_298118 |
| Rat anti-mouse CD34, 1:200 | eBiosciences | Cat# 13-0341, RRID: AB_466424 |
| Rabbit anti-mouse Sox9, 1:1000 | Millipore | Cat# AB5535, RRID: AB_2239761 |
| Rabbit anti-mouse Keratin 15, 1:200 | Abcam | Cat# ab52816, RRID: AB_869863 |
| Mouse anti-mouse alpha smooth muscle actin, 1:200 | Abcam | Cat# ab7817, RRID: AB_262054 |
| Rabbit anti-mouse CD3, 1:200 | Abcam | Cat# ab16669, RRID: AB_443425 |
| Rabbit anti-mouse CD4, 1:200 | Abcam | Cat# ab183685, RRID: AB_2686917 |
| Rabbit anti-mouse CD8, 1:200 | Abcam | Cat# ab217344, RRID: AB_2890649 |
| Goat anti-mouse CD45, 1:200 | Bio-Techne | Cat# AF114, RRID: AB_442146 |
| Rat anti-mouse FOXP3, 1:100 | Thermo Fisher Scientific | Cat# 14-5773-82, RRID: AB_467576 |
| Brilliant Violet 650™ anti-mouse I-A/I-E, 1:200 | Biolegend | Cat#107641, RRID: AB_2565975 |
| PE anti-mouse IFN-γ, 1:100 | Biolegend | Cat# 505807, RRID: AB_315401 |
| Rat Anti-Mouse CD45 (30-F11), PerCp, 1:200 | TonboTM | Cat# 67-0451 |
| PE/Cyanine7 anti-mouse CD335 (NKp46), 1:200 | Biolegend | Cat# 137617, RRID: AB_11218594 |
| Alexa Fluor® 647 anti-mouse IL-17A, 1:100 | Biolegend | Cat# 506911, RRID: AB_536013 |
| Rat Anti-Mouse Ly-6G/Ly-6C (Gr-1) (RB6-8C5), redFluor™ 710, 1:200 | TonboTM | Cat# 80-5931 |
| APC/Fire™ 750 anti-mouse TCR β chain, 1:200 | Biolegend | Cat# 109245, RRID: AB_2629696 |
| FITC anti-mouse NK-1.1 Antibody, 1:200 | Biolegend | Cat# 108705, RRID: AB_313392 |
| PE anti-mouse CD314 (NKG2D), 1:200 | Biolegend | Cat# 115605, RRID: AB_313658 |
| APC Anti-mouse TCR γ/δ, 1:200 | Biolegend | Cat# 118115, RRID: AB_1731824 |
| PE/Cyanine7 anti-mouse I-A/I-E, 1:200 | Biolegend | Cat# 107629, RRID: AB_2290801 |
| Brilliant Violet 650™ anti-mouse CD45, 1:200 | Biolegend | Cat# 103151, RRID: AB_2565884 |
| PE anti-mouse CD326 (Ep-CAM), 1:200 | Biolegend | Cat# 118205, RRID: AB_1134176 |
| PE/Cyanine7 anti-mouse CD326 (Ep-CAM), 1:200 | Biolegend | Cat# 118215, RRID: AB_1236477 |
| FITC Rat anti-Mouse CD34, 1:200 | eBiosciences | Cat# 11-0341-82 RRID: AB_1645242 |
| APC anti-mouse CD200 (OX2), 1:200 | Biolegend | Cat# 123809, RRID: AB_10900996 |
| PE/Cyanine7 anti-mouse Ly-6A/E (Sca-1), 1:200 | Biolegend | Cat# 108113, RRID: AB_493597 |

| Reagent/Resource | Reference or Source | Identifier or Catalog Number |
|---|---|---|
| Alexa Fluor® 700 anti-mouse H-2Kb, 1:100 | Biolegend | Cat# 116521, RRID: AB_2750392 |
| PE anti-mouse CD124 (IL-4Rα), 1:200 | Biolegend | Cat# 144803, RRID: AB_2561729 |
| PE anti-mouse FOXP3, 1:100 | Biolegend | Cat# 126403, RRID: AB_1089118 |
| Ultra-LEAF™ Purified anti-mouse CD16/32 Antibody, 1:1000 | Biolegend | Cat# 101330 |
| anti-NK1.1 (clone PK136) | (Drobits et al, 2012) | N/A |
| InVivoMab anti-CD8α (clone 2.43) | BioXCell | Cat# BE0061 |
| InVivoMab rat IgG2b isotype control (clone LTF-2) | BioXCell | Cat# BE0090 |
| Rabbit polyclonal to STAT1 (phospho Y701) | Abcam | Cat# ab30645 |
| **Oligonucleotides and other sequence-based reagents** | | |
| Egfr fwd: TTGGAATCAATTTTACACCGAAT | Eurofins | N/A |
| Egfr rev: GTTCCCACACAGTGACACCA | Eurofins | N/A |
| H2kb fwd: GTGATCTCTGGCTGTGAAGT | Eurofins | N/A |
| H2kb rev: GTCTCCACAAGCTCCATGTC | Eurofins | N/A |
| Gadph fwd: GGGTTCCTATAAATACGGACTGC | Eurofins | N/A |
| Gadph rev: CCATTTTGTCTACGGGACGA | Eurofins | N/A |
| **Chemicals, Enzymes and other reagents** | | |
| Kefzol (Cefazolin) | Astro Pharma GmbH | Cat# 2453958 |
| Erlotinib, Free Base | Santa Cruz Biotechnology | Cat# sc-396113 |
| Ruxolitinib | LC Labs | Cat# R-6688 |
| DMSO | Sigma-Aldrich | Cat# D8418 |
| Deoxyribonuclease I | Sigma-Aldrich | Cat# DN25 |
| Trypsin (1:250), powder | Thermo Fisher Scientific | Cat# 27250018 |
| Formalin solution, neutral buffered, 10% | Sigma-Aldrich | Cat# HT501128 |
| Recombinant INFγ | Peprotech | Cat#315-05 |
| Recombinant Murine IL-4 | Peprotech | Cat# 214-14 |
| Recombinant Mouse IL-13, CF | Bio-Techne | Cat# 413-ML-005/CF |
| Recombinant Mouse Oncostatin (OSM), CF | Bio-Techne | Cat# 495-MO-025/CF |
| **Critical Commercial Assays** | | |
| Custom Multiplex Cytokine Luminex Assay | Thermo Fisher Scientific | N/A |
| Trizol LS Reagent | Thermo Fisher Scientific | Cat# 11588616 |
| miRNeasy Micro Kit | Qiagen | Cat# 217084 |
| Superscript IV Reverse Transcriptase | Thermo Fisher Scientific | Cat# 18090050 |
| SYTOX™ Blue Dead Cell Stain | Thermo Fisher Scientific | Cat#10297242 |
| Zombie Aqua Fixable Viability Kit | Biolegend | Cat# B423102 |
| eBioscience™ Foxp3/Transcription Factor Staining Buffer Set | Thermo Fisher Scientific | Cat# 00-5523-00 |
| Precellys® Ceramic kit 2.8 mm, 50 × 2 ml tubes, pre-filled with ceramic beads | VWR | Cat# 432-3752 |

| Reagent/Resource | Reference or Source | Identifier or Catalog Number |
|---|---|---|
| Compact Dry Swab | HyServe | Cat# 1 002 953 |
| SignalStain® Boost IHC Detection Reagent (HRP, Rabbit) | Cell Signalling Technologies | Cat# 8114 |
| Power SYBR® Green PCR Master Mix | Thermo Fisher Scientific | Cat# 10209284 |
| Chromium Next GEM Single Cell 5' Kit v2 | 10x Genomics | Cat# 1000265 |
| **Software** | | |
| FlowJo v10 | FlowJo, LLC | https://www.flowjo.com/ |
| GraphPad Prism 8.0.2 | GraphPad Software | https://www.graphpad.com/ |
| xPONENT software 3.1 | Luminex Corp | N/A |
| Adobe Illustrator CS6 | Adobe Inc. | https://www.adobe.com/ |
| Adobe Photoshop Vers13 | Adobe Inc. | https://www.adobe.com/ |
| Halo v3.6.4134.396 and HALO AI 3.6.4134 | Indica Labs | https://indicalab.com/halo/ |
| Definiens Tissue Studio 3.0 | Definiens | https://definiens.zendesk.com/ |
| R 4.3.2 | The Comprehensive R Archive Network | https://cran.r-project.org/ |
| **Other** | | |
| RNA-seq dataset | This study | GSE273571 |
| Single-Cell Seq dataset | This study | GSE273572 |
| Spatial transcriptomics dataset | Cohen et al, 2024 | GSE227632 |

## Mice

EGFR$^{\Delta ep}$ and K5-SOS transgenic mice were generated as previously described in Lichtenberger et al (2013) and Klufa et al (2019). Egr2-Cre mice were purchased from The Jackon Laboratory. EGFR$^{\Delta Egr2}$ were generated by crossing Egr2-Cre with mice carrying conditional EGFR alleles EGFR$^{fl/fl}$. STAT1$^{fl/fl}$ were provided by Assoc. Prof. Priv.-Doz. Mag. Dr. Robert Eferl and generated by Wallner et al (2012). JAK1$^{fl/fl}$ and JAK2$^{fl/fl}$ mice were provided by DI Dr Alexander Dohnal, Univ.-Prof. Dr. med.univ. Veronika Sexl and O.Univ.-Prof. Dr. med.vet. Matthias Müller and Univ.-Prof. Dr. Emilio Casanova. JAK2$^{fl/fl}$ mice were generated as described in Wagner et al (2004) and Krempler et al (2004). EGFR STAT1$^{\Delta Egr2}$ or EGFR JAK1/2$^{\Delta Egr2}$ were generated by crossing EGR2-Cre and EGFR$^{fl/fl}$, STAT1$^{fl/fl}$, JAK1$^{fl/fl}$ and JAK2$^{fl/fl}$, respectively. Conditional knockout mice were kept with food and water ad libitum at the mouse facilities of the Medical University of Vienna and handled according to the standards and regulations approved by the animal experimental ethics committee and the Austrian Ministry of Science and Research (animal license number GZ BMWFW-66.009/0319-V/3b/2019). Corresponding to the Arrive guidelines (Percie du Sert et al, 2020), all experiments were designed to use the smallest number of animals, which are identified in the legends of the Figs. Mice were allocated randomly to experiments and groups independent of sex. In all experimental set-ups, relevant treated and untreated wild type and EGFR$^{\Delta Egr2}$ litter mates served as controls. Mice were treated with depletion recombinant antibodies i.p., topical pharmacological inhibitors and systemic antibiotics using the protocols below.

## TEWL measurement

TEWL of dorsal skin was measured with a Tewameter® TM 300 probe attached to the MDD4-display device (Courage + Khazaka) according to the manufacturer's recommendations.

## Antibiotic treatment

The drinking water of mouse cages was supplemented with cefazolin (0.5 g/L) (Astro Pharma) and replaced twice weekly. For increased cleanliness, cages were changed twice weekly.

## Depletion antibody treatment

For in vivo depletion, mice were administered i.p. twice weekly with 400 μg InVivo*Mab* anti-NK1.1 (clone PK136), 300 μg InVivo*Mab* anti-CD8α (clone 2.43, BioXCell) or InVivo*Mab* rat IgG2b isotype control (clone LTF-2, BioXCell) in respective concentrations for 4 weeks.

## Topical Ruxolitinib treatment

Mice were treated topically daily with 100 μl 3% Ruxolitinib (LC Labs, R-6688) or vehicle control only (DMSO) for 4 weeks.

## Flow cytometric analysis of lymph node single-cell suspensions

Skin-draining lymph nodes were mechanically homogenized and filtered through a 70 μm cell strainer to achieve a single-cell suspension in 2% FBS in PBS. The single-cell suspension was blocked in FC-block CD16/32 (1:100, Biolegend) and subsequently incubated in Zombie Aqua viability stain (1:200, Biolegend for 15 min at room temperature. After another washing step, cells were resuspended in 50 μL staining buffer with the extracellular antibodies (see Table) and incubated for 30 min on ice. Cells were recorded using an LSR-II flow cytometer (BD Biosciences) and analyzed using FlowJo software.

## Fluorescence-activated cell sorting of epidermal single-cell suspensions

Harvested mouse ears were split into the dorsal and ventral sides and incubated in 0.8% trypsin (Gibco, Thermo Fisher Scientific) in PBS at 37 °C for 45 min. Dorsal skin was placed on 0.25% trypsin in DMEM and incubated at 4 °C overnight. The epidermis was separated from the dermis, further, digested in 250 μg/ml DNAseI (Sigma-Aldrich) for 30 min at 37 °C, and washed and filtered using a 70 μm cell strainer.

Single-cell suspensions were subsequently blocked with FC-block CD16/32 (Biolegend) and either stained with extracellular fluorescent antibodies at 4 °C for 30 min and prior flow cytometric analysis SYTOX™ Blue Dead Cell Stain (1:1000, Thermo Fisher Scientific) was added. For intracellular staining of epidermal single-cell suspensions, cells were stained with Zombie Aqua viability stain (1:200, Biolegend) for 15 min at room temperature, following an extracellular antibody staining mix. Next, cells were fixed in 3% Formalin solution (Sigma-Aldrich) in PBS for 10 min and

permeabilized in Perm/Wash buffer (BD, 1:10) for 20 min at room temperature. Fluorescent antibodies for intracellular cytokines were added to Perm/Wash buffer and incubated for 30 min on ice. Cells were recorded using an LSR-II flow cytometer (BD Biosciences) or Cytek™ Aurora and analyzed using FlowJo software.

## Bulk RNA sequencing

Using the protocol above for Fluorescence-activated cell sorting of epidermal single cell suspensions, viable CD45⁻CD34⁺Sca-I⁻ cells were sorted from epidermal single cell suspensions of 1-month-old EGFR$^{\Delta Egr2}$ ($n = 3$) and littermate controls ($n = 3$) into Trizol LS Reagent (Thermo Fisher Scientific) using a FACS Aria III. RNA was extracted with the column-based miRNeasy Micro Kit (Qiagen) according to the manufacturer's protocol. cDNA synthesis was performed with Superscript IV Reverse Transcriptase (Thermo Fisher Scientific). The main targets were confirmed by real-time PCR using SYBR-Green (Thermo Fisher Scientific). Samples were sent to Novogene, UK for quality control, library preparation and sequencing using Novaseq PE150. The kallisto pipeline was used to quantify the counts from the raw sequencing data (Bray et al, 2016). Differential gene expression analysis was performed using DESeq2 and Gene Set Enrichment Analysis (Love et al, 2014).

## Single-cell RNA sequencing

Using the protocol above for Fluorescence-activated cell sorting of epidermal single cell suspensions, 12,000 viable CD45⁻Sca-I⁻ cells and 12,000 viable CD45⁺ were sorted from pooled epidermal single cell suspensions of 3-month-old EGFR$^{\Delta Egr2}$ ($n = 4$) into a 0.04% BSA solution using a BD FACSMelody™ Cell Sorter (Biosciences). Single-cell cDNA libraries were generated using the droplet-based Chromium Next GEM Single Cell 5' Kit v2 (Cat.nr. 1000265, 10x Genomics) following manufacturer's instructions. Libraries were sequenced on insert machine name. The CellRanger pipeline version 7.0.0 was used to process raw sequencing files and produce feature-barcode matrices which were further analyzed in using several R packages. Most analyses were carried out using functions from the Seurat package (Hao et al, 2021) unless otherwise noted. First, ambient RNA contamination, which is commonly observed in droplet-based scRNA-seq methods, was corrected by using the SoupX package (Young and Behjati, 2020). Corrected count matrices were then imported into the standard Seurat workflow which includes initial quality control steps and filtering, cell cycle scoring, normalization, dimensionality reduction, clustering, and visualization. Doublets were identified using the scDblFinder package (Germain, 2022). For the downstream analysis, we included 5291 cells which passed quality control parameters (singlets, number of genes ≥300, number of UMIs ≥500, percentage of mitochondrial reads <5%). Cell cycle scoring and the difference between G2M and S phase scores ("CC.Difference") were calculated. Normalization was carried out using the SCTransform function with CC. Difference and percentage of mitochondrial reads set as variables to regress. For dimensionality reduction and visualization purposes, we used the RunPCA and RunUMAP functions of Seurat. For clustering and cluster marker identification, we used Seurat's FindNeighbours, FindClusters and FindAll-Markers functions. Clusters were manually annotated using

published single-cell datasets of mouse skin (Joost et al, 2016). Predicted ligand-receptor interactions were determined with the CellChat package (Jin et al, 2021).

## Ex vivo skin explant

Ex vivo skin explants were prepared as described previously (Bauer et al, 2021). Fresh mouse ears were split and floated dermal side down on 1 ml RPMI medium containing 10% FBS and in a 24-well plate at 37 °C for 48 h (Bauer et al, 2021). Medium was supplemented with 120 ng/mL INFγ, 20 ng/mL IL-4 (Preprotech), 20 ng/mL OSM or 20 ng/mL IL-13 (Bio-Techne). Analysis was performed using the protocol above for fluorescence-activated cell sorting of epidermal single-cell suspensions.

## Keratinocyte culture

Epidermal single-cell suspensions were isolated as previously described for fluorescence activated cell sorting and cultured on fibronectin-coated dishes in keratinocytes growth medium 2 (Promo2) supplemented with 0.02 mM $CaCl_2$ (Promo Cell), growth supplements (1:40), and Penicillin/Streptavidin (1:100, Sigma). 80% confluent cells were treated with Erlotinib (10 mM; Santa Cruz Biotechnology) for 24 h. Next, the medium was supplemented with 120 ng/mL INFγ (Peprotech) and 400 nM Ruxolitinib (LC Labs, R-6688).

## Full skin protein lysates and Luminex

Snap-frozen dorsal skin was added to RIPA lysis buffer, supplemented with Protease Inhibitor Cocktail (Roche) and homogenized using a Precellys 24 homogenizer (Bertin). Skin lysates were centrifuged at $14,000 \times g$ for 15 min at 4 °C to remove cell debris. For protein quantification, lysates were thawed on ice and subjected to Bradford protein quantification, according to the manufacturer's protocol (Biorad). 70–100 μg total protein was used for each assay. Multiplex Luminex assays (Thermo Fisher Scientific) were performed according to the manufacturer's recommendations and measured on a Luminex MAGPIX System using the xPONENT Software.

## Epidermal sheets

Epidermal sheets were prepared as described previously (Klufa et al, 2019). Briefly, mouse tail sheets were floated on 3.8% ammonium thiocyanate for 25 min and fixed with 4% para-formaldehyde (PFA) for 30 min at room temperature before immunofluorescence staining.

## Histological immunofluorescence and immunohistochemistry analysis

Dorsal skin was fixed in 4% PFA, embedded in a paraffin block and cut into 4 μm sections. After dewaxing and rehydration, sections were stained with hematoxylin and eosin (H&E), according to standard procedures of Papanicolaou. For immunofluorescence stainings, skin samples were heated in antigen retrieval solution using Dako Target Retrieval Solution (dilution 1:10, pH = 6 or pH = 9). For immunohistochemistry, samples were treated with 3%

$H_2O_2$ before blocking. All samples were blocked with 5% horse or goat serum in 2% BSA TBS-T for 1 h in a humidified slide chamber at room temperature and stained with primary antibodies at 4 °C overnight. The next day, slides were rinsed and incubated with an appropriate secondary fluorescence antibody (1:400) and Hoechst (Sigma-Aldrich) or with Signal stain Boost IHC Detection reagent (HRP) for 2 h in a dark humidified slide chamber. Tissue sections were mounted, and immunofluorescence pictures were taken using a Nikon Eclipse i80 microscope. For immunohistochemistry, skin tissues was stained with a DAB kit and hematoxylin. Slides were scanned and analyzed using Definiense or Halo software.

## Human skin specimens and proof-of-concept treatment

All experiments conform to the principles set out in the WMA Declaration of Helsinki and the Department of Health and Human Services Belmont Report. Paraffin-embedded biopsy samples from folliculitis decalvans ($n = 7$), frontal fibrosing alopecia ($n = 8$) and lichen planopilaris ($n = 5$) patients were obtained from the biobanks of the Departments of Dermatology at the University Hospital Düsseldorf and the Medical University of Vienna (Ethics approval ID 2016075402, Heinrich-Heine University, 40225 Duesseldorf, Germany; Ethics approval 1354/2021, Medical University of Vienna). In addition, skin biopsies were available from two patients from the Department of Dermatology Rudolfstiftung Hospital (Vienna, Austria) who received cetuximab for the treatment of inoperable SCC and gave consent for the retrospective use of their data and samples. See more details in Klufa et al (Klufa et al, 2019).

## Spatial transcriptomics

Published Visium V1 data from Cohen et al (2024) (Data Ref: Cohen et al, 2024) were used for the spatial transcriptomic analysis of human LPP. Already processed spaceranger output files were downloaded from GEO with the accession number GSE227632. In addition, tissue_positions_list.csv, scalefactors_json.json files and a high resolution H&E image were provided by authors. The data included 2 healthy and 2 scarring alopecia tissue sections on one Visium V1 capture area. Data analysis was carried out in R version 4.3.2 and using the Seurat package version 5.0.3. and closely followed the Seurat vignette for processing Visium datasets (Hao et al, 2024). Briefly, the Seurat object was created using Read10X and Read10X_Image functions followed by normalization and variance stabilization with SCTransform. Gene expression was then plotted with the SpatialFeaturePlot function.

## Statistics

Statistical analyses were performed with GraphPad Prism 8.0 software. Data were tested for normal distribution by the Shapiro–Wilk test. In cases of normal distribution Student's unpaired two-tailed t-test for comparisons of two groups or parametric One-way ANOVA analysis with Tukey's pairwise comparisons were used to compare more than two groups. Experiments were repeated independently at least two times with similar results. Dot plots depict biological replicates unless otherwise stated (*$P < 0.05$, **$P < 0.01$, ***$P < 0.001$, and ****$P < 0.0001$). The data are shown as mean ± SEM. All experiments were conducted non-blindly, and we included all samples in our analyses.

**The paper explained**

**Problem**

Prolonged inflammation during pathogen removal can lead to host tissue destruction. Evolutionary important body structures such as the hair follicles are protected from inflammatory damage by the immune privilege. However, during cicatricial or scarring alopecia the permanent destruction of the hair follicles is inevitable. This stigmatizing medical condition lacks effective mechanism-based therapeutic treatment options.

**Results**

We identified that prolonged dysfunctional EGFR signaling in the hair follicle leads to scarring hair follicle destruction. Transcriptional profiling of the hair follicle stem cells during alopecia onset revealed a hyper-activated JAK-STAT1 signaling cascade and the upregulation of the antigen presentation machinery. This harmful condition is sustained by interferon-gamma-expressing natural killer cells and cytotoxic T cells. Our study found that blocking the JAK pathway using inhibitors can prevent and reverse this destructive process.

**Impact**

Our research provides mechanistic insights behind scarring hair follicle destruction, particularly relevant for patients undergoing EGFR-inhibitor anti-cancer therapy and patients with cicatricial alopecia. It also presents JAK inhibitors as a potential therapeutic strategy to prevent or treat these permanent forms of alopecia.

## Data availability

RNA-seq datasets are publicly available on Gene Expression Omnibus (GEO) database under RNA-Seq data: Gene Expression Omnibus GSE273571. scRNA-Seq data: Gene Expression Omnibus GSE273572. Source data for Fig. 4 is available at Biostudies BSST1687. The source data of this paper are collected in the following database record: biostudies:S-SCDT-10_1038-S44321-024-00166-3.

## Peer review information

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

## Acknowledgements

We thank Sophie Huszarek and Lukas Ginzinger for excellent technical assistance. We thank Temenuschka Baykuscheva-Gentscheva for help with histology and Johannes Reisecker for his assistance in FACS sorting. We thank the Facility Microscopy and Imaging, Gerald Timelthaler and Dominik Kirchhofer for their assistance in Imaging. The Core Facility Genomics at the Medical University of Vienna is acknowledged for carrying out the RNA sequencing analysis. We are grateful to Martina Hammer and the staff of the Department of Biomedical Research of the Medical University of Vienna for maintaining our mouse colonies. This work was supported by grants from the Austrian Science Fund (FWF: I4300 and PAT5056523 to T Bauer; PhD program W1212 "Inflammation and Immunity" to M Sibilia). M Sibilia's research is funded by the WWTF and the European Research Council (ERC) grant (ERC-2015-AdG TNT-Tumors 694883). The graphical abstract was created with Biorender.com.

## Author contributions

**Karoline Strobl**: Conceptualization; Data curation; Formal analysis; Validation; Investigation; Visualization; Methodology; Writing—original draft; Writing—review and editing. **Jörg Klufa**: Conceptualization; Formal analysis; Validation; Investigation; Methodology. **Regina Jin**: Formal analysis; Methodology. **Lena Artner-Gent**: Methodology. **Dana Krauß**: Formal analysis. **Philipp Novoszel**: Methodology. **Johanna Strobl**: Methodology. **Georg Stary**: Resources; Methodology. **Igor Vujic**: Resources; Methodology. **Johannes Griss**: Resources; Methodology. **Martin Holcmann**: Methodology. **Matthias Farlik**: Methodology. **Bernhard Homey**: Conceptualization; Resources; Methodology. **Maria Sibilia**: Resources; Supervision; Funding acquisition; Project administration; Writing—review and editing. **Thomas Bauer**: Conceptualization; Resources; Data curation; Formal analysis; Supervision; Funding acquisition; Validation; Investigation; Visualization; Methodology; Writing—original draft; Project administration; Writing—review and editing.

Source data underlying figure panels in this paper may have individual authorship assigned. Where available, figure panel/source data authorship is listed in the following database record: biostudies:S-SCDT-10_1038-S44321-024-00166-3.

## Disclosure and competing interests statement

The authors declare no competing interests.

# Expanded View Figures

**Figure EV1.  Hair follicle-specific EGFR deletion induces epidermal immune infiltrate and microbiota-driven inflammatory hair follicle destruction.** ▶

(**A**) Representative pictures of the gating strategy of FACS analysis of epidermal single cell suspensions (WT and EGFR$^{\Delta Egr2}$ mice). CD34$^+$ Sca-I$^-$ HFSC, CD34$^-$ Sca-I$^-$ HF and Sca-I$^+$ IFE among CD45$^-$ keratinocytes. γδTCR$^{hi}$ dendritic epidermal T cells (DETC), γδTCR$^{int}$ γδT cells (γδTC), αβT cells (αβTC), CD11b$^+$Gr-1$^+$ neutrophils, NK1.1$^+$ Nkp46$^+$ natural killer (NK) cells, CD11b$^+$Epcam$^+$ Langerhans cells (LC) among CD45$^+$ immune cells. (**B**) FACS analysis of αβT cells (WT vs EGFR$^{\Delta ep}$ $p = 0.0067$, EGFR$^{\Delta ep}$ vs WT Abx $p = 0.001$, EGFR$^{\Delta ep}$ vs EGFR$^{\Delta ep}$ K5-SOS Abx $p = 0.0057$) and γδT cells (WT vs EGFR$^{\Delta ep}$ K5-SOSp <0.0001, EGFR$^{\Delta ep}$ vs EGFR$^{\Delta ep}$ K5-SOS $p = 0.013$, EGFR$^{\Delta ep}$ K5-SOS vs WT Abx $p < 0.0001$, EGFR$^{\Delta ep}$ K5-SOS vs EGFR$^{\Delta ep}$ Abx $p = 0.0026$, EGFR$^{\Delta ep}$ K5-SOS vs EGFR$^{\Delta ep}$ K5-SOS Abx $p = 0.0004$) among CD45$^+$ immune cells at 2 M. Each dot represents an independent mouse. Mouse models as indicated in the graph. (**C**) Kaplan–Meier survival plot of EGFR$^{\Delta Egr2}$ mice or WT. (**D**) Quantification of EGFR immunohistochemistry staining by Definiens software. Percentage of EGFR expression in hair follicle ($p = 0.0165$) and epidermis of EGFR$^{\Delta Egr2}$ mice or WT at 3 M. (**E**) Representative pictures of hair follicle length measured from hematoxylin and eosin (H&E) stained skin sections marked in yellow dotted lines of WT or EGFR$^{\Delta Egr2}$. (**F**) Sox9 (red) and CD34 (green) positive stem cells of WT and EGFR$^{\Delta Egr2}$ mice at 2, 3, and 5 months. Quantified by Halo AI software in percentage of the hair follicle (CD34$^+$ $p = 0.0227$, Sox9$^+$ $p = 0.0055$). (**G**) Timeline P8-P150 showing representative pictures of H&E stainings of WT and EGFR$^{\Delta Egr2}$. Data is presented in ±SEM, *$p < 0.05$, **$p < 0.01$, ***$p < 0.001$, ****$p < 0.0001$ by One-Way ANOVA with Tukey's posthoc correction, $n \geq 3$.

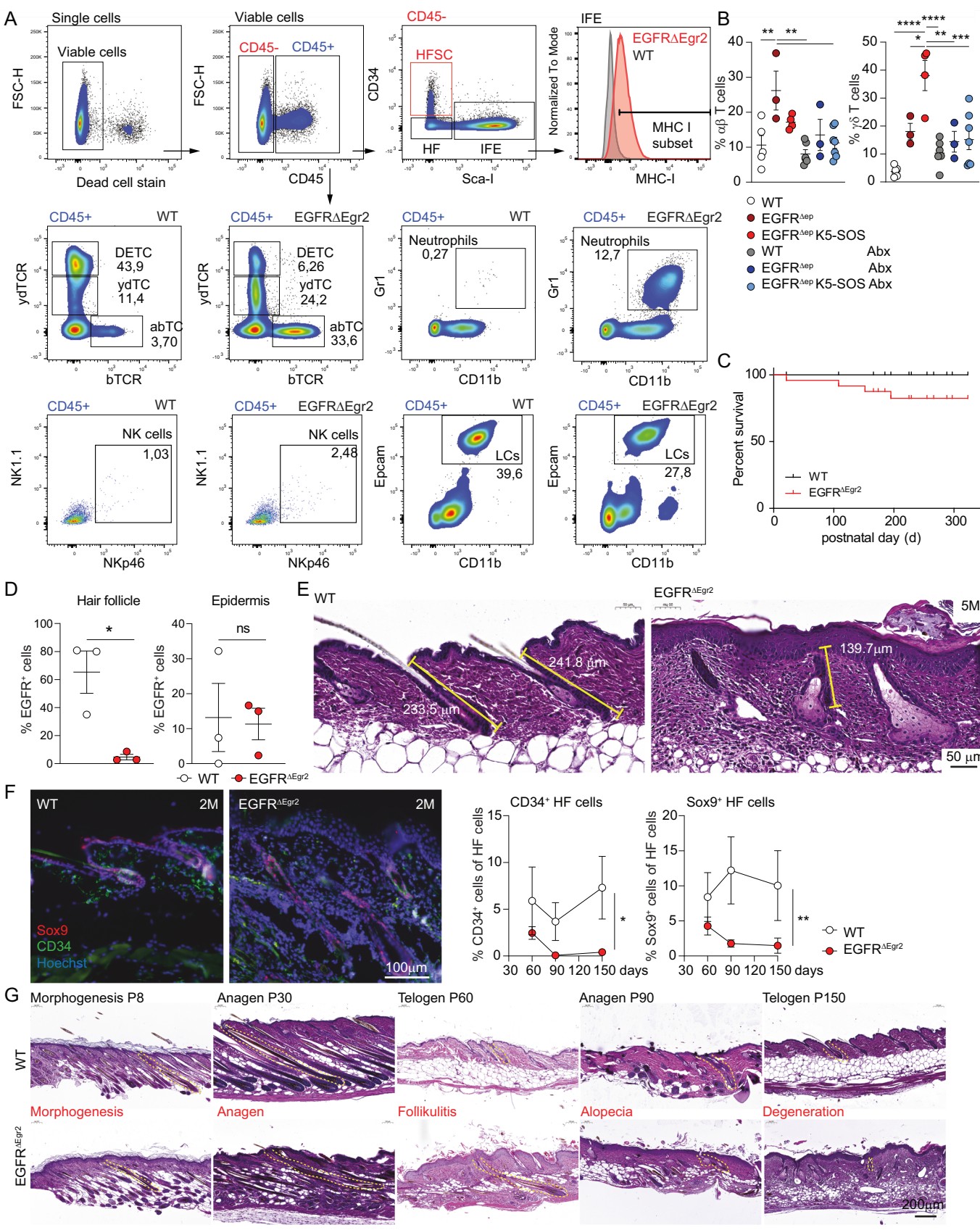

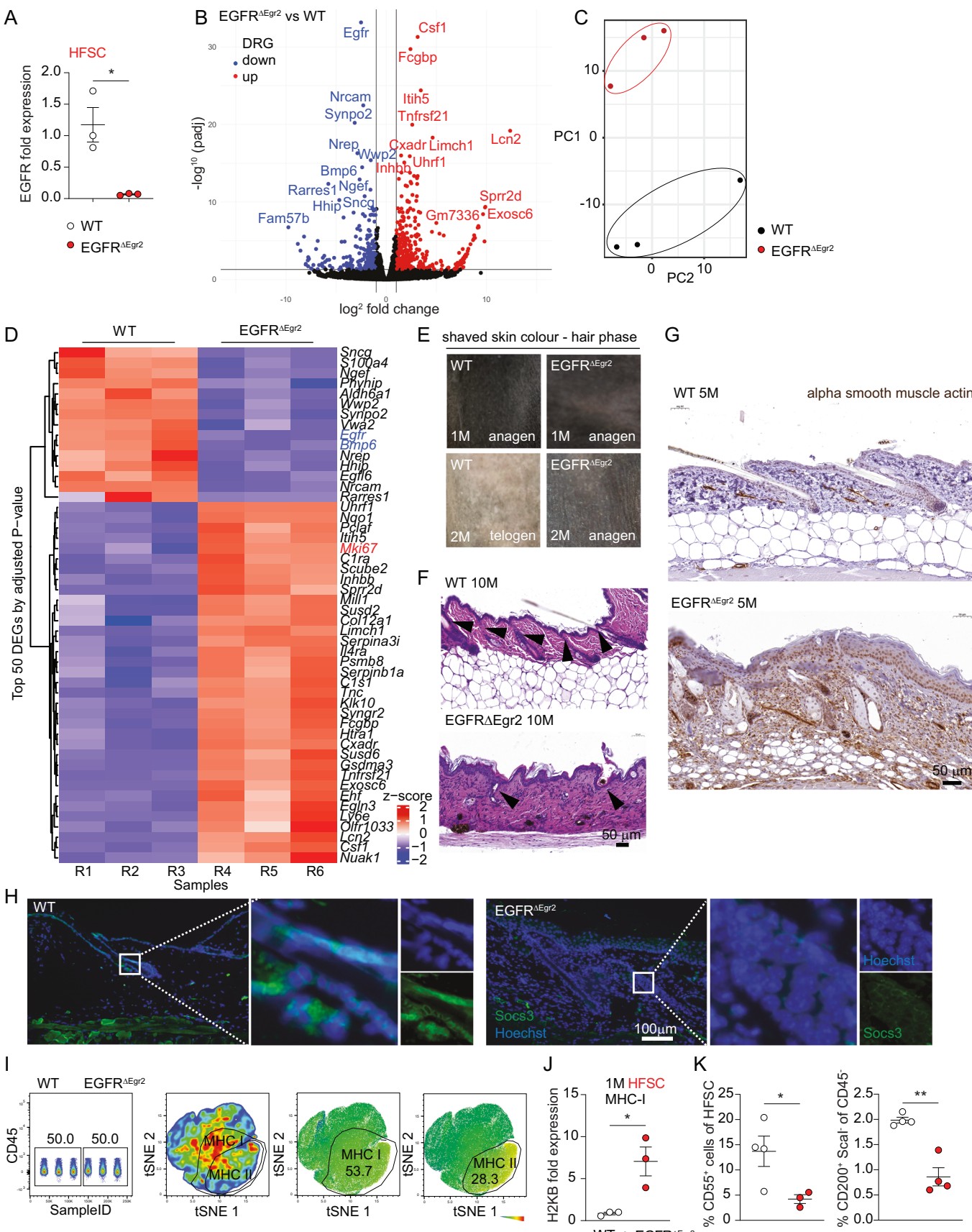

◄ **Figure EV2. RNA profiling of FACS sorted EGFR-deficient hair follicle stem cells before scarring hair follicle destruction.**

(A) Real-time PCR of EGFR expression of sorted CD34$^+$Sca1$^-$ HFSCs of WT and EGFR$^{\Delta Egr2}$ mice at 1 M. Data is presented in ±SEM, $p = 0.0156$ by unpaired t-test, $n \geq 3$. (B) Volcano plot of differentially expressed genes (blue downregulated, red upregulated) of RNA sequencing analysis of CD34$^+$ HFSCs from WT vs EGFR$^{\Delta Egr2}$ mice. Data shown as fold change (log2) and $p$-value ($-$log10). DESeq2 including local dispersion estimation and Independent Hypothesis Weighting (IHW) (x) to estimate false discovery rates and power maximization were used for statistical analysis (Ignatiadis et al, 2016). (C) Principal component (PC) analysis of the RNAseq dataset. (D) Heatmap of z-scores of top 50 differentially expressed genes by adjusted $p$-value of the RNAseq dataset. Genes of special interest are marked in red (up) or blue (down). Statistics DESeq2 as in (B). (E) Pictures of shaved backs of WT or EGFR$^{\Delta Egr2}$ at 1 M and 2 M of age. (F) Representative H&E stained skin sections for counting the number of hair follicle units per 1500 μm of WT and EGFR$^{\Delta Egr2}$ skin at 10 M of age. Black arrowheads indicate counted hair follicles. (G) Representative alpha smooth muscle actin staining on skin sections of WT and EGFR$^{\Delta Egr2}$ skin at 5 M of age. (H) Immunofluorescence staining of SOCS3 in green of WT and EGFR$^{\Delta Egr2}$ mouse skin sections at 3 M. (I) tSNE FACS gating strategy of MHC-I and MHC-II expression among CD45$^-$ keratinocytes of WT or EGFR$^{\Delta Egr2}$ at 5 M of age. (J) Real-time PCR of MHC-I expression (H-2Kb) of sorted CD34$^+$ Sca1$^-$ HFSCs of WT or EGFR$^{\Delta Egr2}$ mice at 1 M of age ($p = 0.0229$). (K) CD55 ($p = 0.0481$) and CD200 ($p = 0.001$) surface expression in HFSCs and HF of WT and EGFR$^{\Delta Egr2}$ mice by FACS analysis. Data is presented in ±SEM, *$p < 0.05$, **$p < 0.01$ by unpaired t-test, $n \geq 3$.

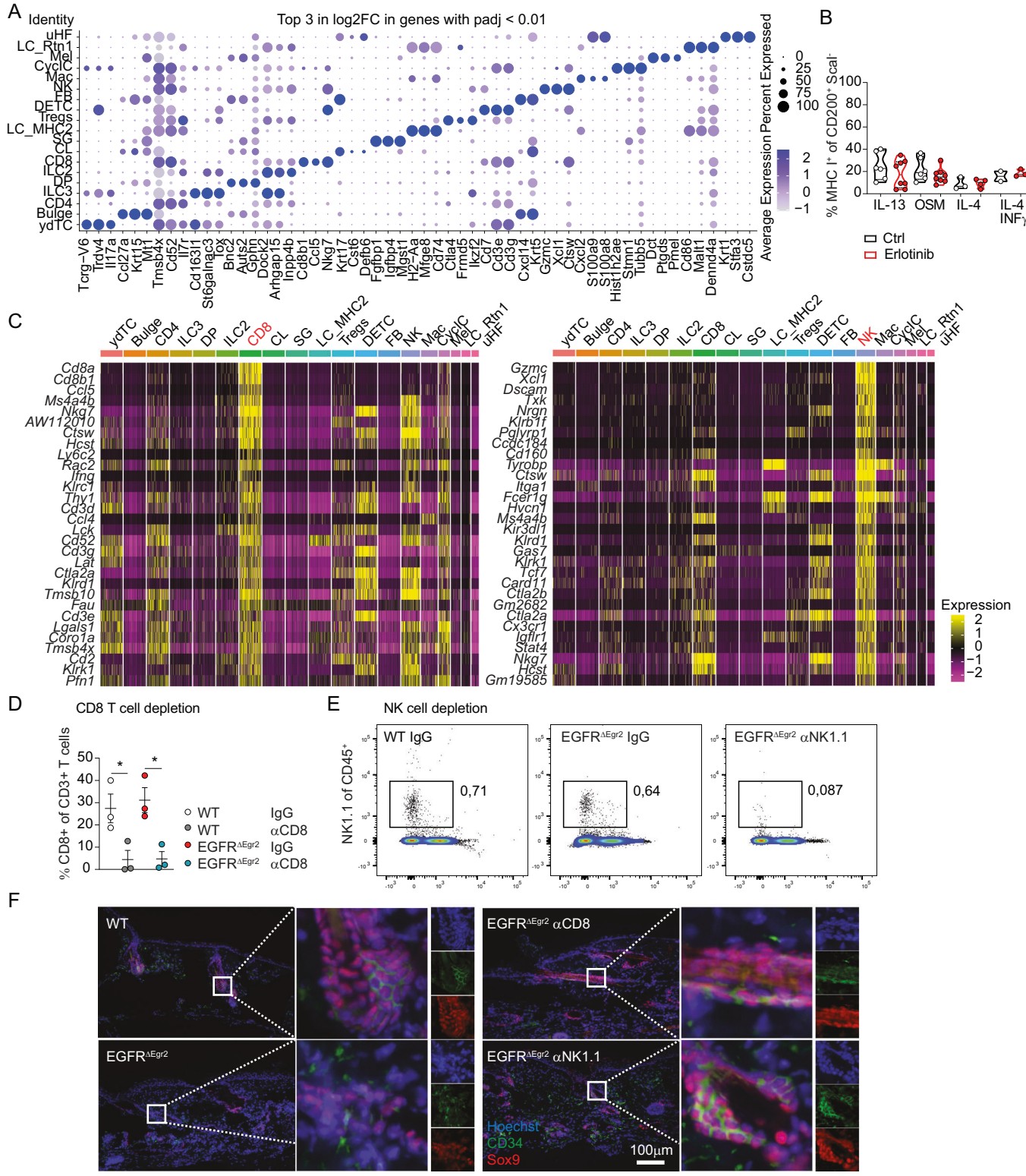

◀ **Figure EV3.  Single-cell analysis identified hair follicle and immune cell populations in EGFR^ΔEgr2 mouse epidermal cell suspensions and cytokine impact of HF specific MHC expression.**

(A) Top 3 differentially expressed genes of every cluster of single-cell RNA sequencing analysis of epidermal CD45+ immune cells and CD45- Sca-I- hair follicle cells in EGFR^ΔEgr2. Wilcoxon rank sum test and Bonferroni correction for statistical analysis. (B) Ex vivo WT skin explants treated with indicated cytokines with or without erlotinib for 48 h. FACS analysis of MHC-I and -II expressions in CD200+ HF cells. (C) Top 30 differentially expressed genes of CD8 and NK cell cluster. (D) Confirmation of CD8 T cell (WT $p = 0.0481$, EGFR^ΔEgr2 $p = 0.0246$) and (E) NK cell depletion by FACS analysis of EGFR^ΔEgr2 mice and their respective controls. (F) IF staining of CD34+ and Sox9+ stem cells in skin sections of EGFR^ΔEgr2 treated with the indicated depletion antibodies and the respective controls. Data is presented in ±SEM, *$p < 0.05$ by One-Way ANOVA with Tukey's posthoc correction, $n \geq 3$.

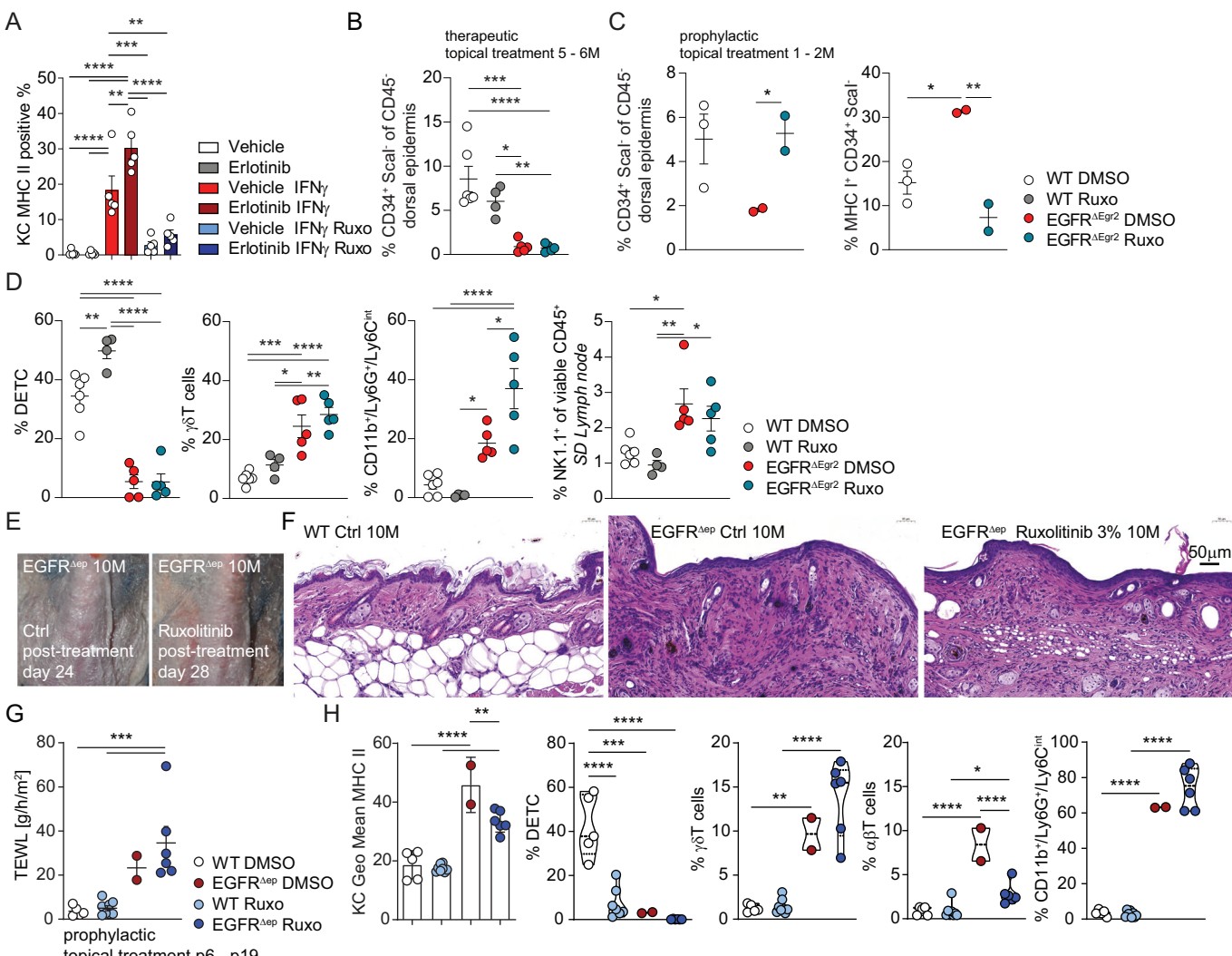

**Figure EV4. Therapeutic and prophylactic JAK inhibition ameliorates skin inflammation in EGFR$^{\Delta Egr2}$ and EGFR$^{\Delta ep}$ mice.**

(A) MHC-II expression of in vitro primary murine KCs of WT mice treated with IFNγ and JAK1/2 inhibitor (ruxolitinib) with or without the EGFR-inhibitor erlotinib (Veh vs Veh IFNγ, Veh vs Erlotinib INFγ, Erlotinib vs Veh IFNγ and Erlotinib vs Erlotinib INFγ $p < 0.0001$, Veh IFNγ vs Erlotinib INFγ $p = 0.0085$, Veh INGγ vs Veh IFNγ Ruxo $p = 0.0005$, Veh INFγ vs Erlotinib IFNγ Ruxo p = 0.0054, Erlotinib INFγ vs Veh INFγ Ruxo and Erlotinib INFγ vs Erlotinib INFγ Ruxo $p < 0.0001$). (B) Summary of FACS analysis of CD34$^+$ hair follicle stem cells from 5 M old WT and EGFR$^{\Delta Egr2}$ mice treated therapeutically with DMSO or ruxolitinib in DMSO for 1 month (WT vs EGFR$^{\Delta Egr2}$ $p = 0.0001$, WT vs EGFR$^{\Delta Egr2}$ Ruxo $p < 0.0001$, EGFR$^{\Delta Egr2}$ vs WT Ruxo $p = 0.0121$, WT Ruxo vs EGFR$^{\Delta Egr2}$ Ruxo $p = 0.0098$). (C) FACS analysis of HFSC ($p = 0.0493$) and MHC-I expression (WT vs EGFR$^{\Delta Egr2}$ $p = 0.0224$, EGFR$^{\Delta Egr2}$ vs EGFR$^{\Delta Egr2}$ Ruxo $p = 0.0076$) on EGFR$^{\Delta Egr2}$ and WT mice treated prophylactically with 3% ruxolitinib from 1 to 2 M of age. (D) FACS analysis of epidermal CD45$^+$ immune cells of EGFR$^{\Delta Egr2}$ and WT treated therapeutically with 3% Ruxolitinib from 5 M to 6 M (DETC: WT vs WT Ruxo $p = 0.0088$, other $p < 0.0001$, γδT cells: WT vs EGFR$^{\Delta Egr2}$ $p = 0.0004$, WT vs EGFR$^{\Delta Egr2}$ Ruxo $p < 0.0001$, EGFR$^{\Delta Egr2}$ vs WT Ruxo $p = 0.0115$, WT Ruxo vs EGFR$^{\Delta Egr2}$ $p = 0.0012$, CD11b/Ly6G$^+$/Ly6C$^{int}$: WT vs EGFR$^{\Delta Egr2}$ Ruxo and WT Ruxo vs EGFR$^{\Delta Egr2}$ Ruxo $p < 0.0001$, EGFR$^{\Delta Egr2}$ WT Ruxo $p = 0.0265$, EGFR$^{\Delta Egr2}$ vs EGFR$^{\Delta Egr2}$ Ruxo $p = 0.0131$, NK1.1: WT vs EGFR$^{\Delta Egr2}$ $p = 0.0142$, EGFR$^{\Delta Egr2}$ vs WT Ruxo $p = 0.0056$, WT Ruxo vs EGFR$^{\Delta Egr2}$ Ruxo $p = 0.0381$). (E, F) Representative pictures of the skin (E) and H&E satinings (F) of WT, EGFR$^{\Delta Egr2}$ mice treated topically with DMSO (vehicle ctrl) or 3% ruxolitinib in DMSO daily at the age of 10 M for 4 weeks. (G, H) WT and EGFR$^{\Delta ep}$ mice were treated prophylactically with 3% Ruxolitinib or DMSO starting from P6 until P19 and TEWL (WT vs EGFR$^{\Delta Egr2}$ Ruxo $p = 0.0007$, WT Ruxo vs EGFR$^{\Delta Egr2}$ Ruxo $p = 0.0003$) was measured from the back-skin (G). FACS analysis of these mice for inflammatory parameters (KC MHC II: EGFR$^{\Delta Egr2}$ vs EGFR$^{\Delta Egr2}$ Ruxo $p = 0.0056$, other $p$ values $< 0.0001$, DETC: all $p$ values $< 0.0001$, γδT cells: WT vs EGFR$^{\Delta Egr2}$ $p = 0.0036$, EGFR$^{\Delta Egr2}$ vs WT Ruxo $p = 0.0027$, other $p < 0.0001$, αβT cells: WT vs EGFR$^{\Delta Egr2}$ Ruxo $p = 0.0429$, WT Ruxo vs EGFR$^{\Delta Egr2}$ Ruxo $p = 0.0191$, other $p < 0.0001$, CD11b/Ly6G$^+$/Ly6C$^{int}$: all $p$ values $< 0.0001$) as indicated (H). Data is presented in ±SEM, *$p < 0.05$, **$p < 0.01$, ***$p < 0.00$, ****$p < 0.0001$ by one-way ANOVA with Tukey's posthoc correction, $n \geq 3$.

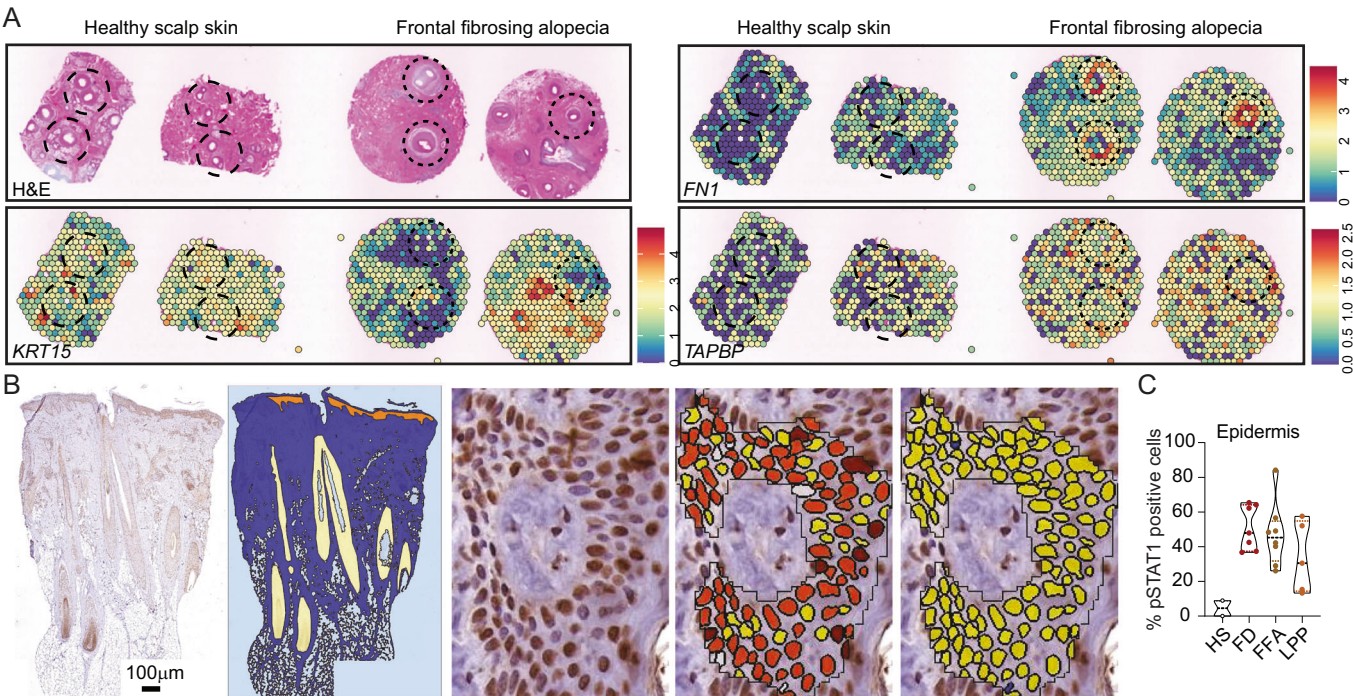

**Figure EV5.  Key features of scarring alopecia in spatial transcriptomics and quantification of phosphorylated STAT1 in human clinical samples.**

(A) Spatial transcriptomic feature plots of the indicated genes superimposed on the corresponding H&E images. The H&E image is also available at Fig. 6C. Dataset from Cohen et al (Data ref: Cohen et al, 2024). (B) Nuclear staining intensity was quantified using definiens software as indicated. Hair follicles (yellow area) and epidermis (orange area) were separately analyzed. Medium intensity is shown in red. (C) Quantification of pSTAT1 staining intensity in the epidermis, $n \geq 2$.

