## [Peer Review File · EMBO Molecular Medicine]

JAK-STAT1 as therapeutic target for EGFR deficiency-associated inflammation and scarring alopecia

Karoline Strobl, Jörg Klufa, Regina Jin, Lena Artner-Gent, Dana Krauss, Philipp Novoszel, Johanna Strobl, Georg Sary, Igor Vujic, Johannes Griss, Martin Holcman, Matthias Farlik, Bernhard Homey, Maria Sibilias, and Thomas Bauer

Corresponding authors: Thomas Bauer (thomas.bauer@meduniwien.ac.at) , Maria Sibilias (maria.sibilias@meduniwien.ac.at)

Review Timeline:

Submission Date:	12th Jun 24
Editorial Decision:	2nd Jul 24
Revision Received:	11th Sep 24
Editorial Decision:	9th Oct 24
Revision Received:	18th Oct 24
Accepted:	24th Oct 24

Editor: Lise Roth

Transaction Report:

2nd Jul 2024

Dear Dr. Bauer,

Thank you for the submission of your manuscript to EMBO Molecular Medicine. We have now received feedback from the three reviewers who agreed to evaluate your manuscript. As you will see from the reports below, the referees acknowledge the interest of the study and are overall supporting publication of your work pending appropriate revisions.

Addressing the reviewers' concerns in full will be necessary for further considering the manuscript in our journal, and acceptance of the manuscript will entail a second round of review.

Regarding the human data (comment from referee #2), and upon further discussion with the referees, we would suggest removing only the 1 patient case study (Figure 6E-F). Referee #2 further commented that the IFITM3 staining panel was unconvincing and should be removed as well unless you can provide more convincing staining. The rest of the human data in Fig. 6 should remain in the manuscript.

EMBO Molecular Medicine encourages a single round of revision only and therefore, acceptance or rejection of the manuscript will depend on the completeness of your responses included in the next, final version of the manuscript. For this reason, and to save you from any frustrations in the end, I would strongly advise against returning an incomplete revision.

We are expecting your revised manuscript within three months, if you anticipate any delay, please contact us.

We require:

4) A .docx formatted letter INCLUDING the reviewers' reports and your detailed point-by-point responses to their comments. As part of the EMBO Press transparent editorial process, the point-by-point response is part of the Review Process File (RPF), which will be published alongside your paper.

5) A complete author checklist, which you can download from our author guidelines (<https://www.embopress.org/page/journal/17574684/authorguide#submissionofrevisions>). Please insert information in the checklist that is also reflected in the manuscript. The completed author checklist will also be part of the RPF.

6) All Materials and Methods need to be described in the main text using our 'Structured Methods' format, which is required for all research articles. According to this format, the Methods section includes a Reagents and Tools Table (listing key reagents, experimental models, software and relevant equipment and including their sources and relevant identifiers) followed by a Methods and Protocols section describing the methods using a step-by-step protocol format. The aim is to facilitate adoption of the methodologies across labs. More information on how to adhere to this format as well as a downloadable template (.docx) for the Reagents and Tools Table can be found in our author guidelines:
<https://www.embopress.org/page/journal/17574684/authorguide#structuredmethods>

7) It is mandatory to include a 'Data Availability' section after the Materials and Methods. Before submitting your revision, primary datasets produced in this study need to be deposited in an appropriate public database, and the accession numbers and database listed under 'Data Availability'. Please remember to provide a reviewer password if the datasets are not yet public (see <https://www.embopress.org/page/journal/17574684/authorguide#dataavailability>).

In case you have no data that requires deposition in a public database, please state so in this section. Note that the Data

Availability Section is restricted to new primary data that are part of this study.

8) For data quantification: please specify the name of the statistical test used to generate error bars and P values, the number (n) of independent experiments (specify technical or biological replicates) underlying each data point and the test used to calculate p-values in each figure legend. The figure legends should contain a basic description of n, P and the test applied. Graphs must include a description of the bars and the error bars (s.d., s.e.m.). Please provide exact p values.

13) Author contributions: CRediT has replaced the traditional author contributions section because it offers a systematic machine readable author contributions format that allows for more effective research assessment. Please remove the Authors Contributions from the manuscript and use the free text boxes beneath each contributing author's name in our system to add specific details on the author's contribution. More information is available in our guide to authors.

16) As part of the EMBO Publications transparent editorial process initiative (see our Editorial at <http://embomolmed.embopress.org/content/2/9/329>), EMBO Molecular Medicine will publish online a Review Process File (RPF) to accompany accepted manuscripts.

In the event of acceptance, this file will be published in conjunction with your paper and will include the anonymous referee reports, your point-by-point response and all pertinent correspondence relating to the manuscript. Let us know whether you agree with the publication of the RPF and as here, if you want to remove or not any figures from it prior to publication.

I look forward to receiving your revised manuscript.

Yours sincerely,

Lise Roth
Lise Roth, PhD
Senior Editor
EMBO Molecular Medicine

***** Reviewer's comments *****

Referee #1 (Comments on Novelty/Model System for Author):

Ideally, the raw data may be made available to reviewers prior during the review process.

Referee #1 (Remarks for Author):

The article EMM-2024-20141 describes a hair follicle specific EGFR knockout model that develops chronic scaling alopecia. Folliculitis is a side effect often seen in patients treated with EGFR inhibitor anticancer therapy. The manuscript contains a comprehensive and rigorous body of work. The results are interesting with clinical relevance and are well-presented. Below are a few minor comments:

1. Is there direct evidence of microbial infection or dysbiosis in the EGFRdepi and EGFRdEGR2 skin? If yes, is there microbial species/strain information?
2. Double knockouts of EGFR and JAK2/STAT1 verified a HFSC-intrinsic JAK2/STAT1 signaling dysregulation as a key driver of folliculitis. In the animal models, JAK2/STAT1 inhibitions in young animals prevented development of folliculitis and restored hair growth in the EGFR-KO skin in 1-2M old mice. Did JAK inhibitor treatment improve hair regrowth in 5-6M old EGFRKO mice?
 - Figure S5D: "Interestingly, JAK1/2 inhibition did not influence the T cell compartment, the neutrophil recruitment and the NK cells (Supplemental Figure 5D)." This experiment was done with 5-6M old mice, but there was no mention about the effects of Ruxolitinib on HFSC in these old mice.
 - Figure S5C. P-value information is missing.
 - Hair regrowth was not apparent in the clinical case of folliculitis decalvans 4-months after JAK inhibitor therapy. How much is known about the etiology for this patient? Does one expect to see hair growth in this patient after a longer period (e.g. 1 year) of treatment?
 - JAK inhibitors are already used for various conditions of alopecia. Does the result of this study imply the importance of early treatment in scarring alopecia?
3. The animal number was provided for bulk RNA-seq (n=3, WT and mutant) and scRNA-se for 3-month-old EGFR Δ Egr2 (n=4). Is WT skin included for scRNA? The animal number information is not clear for other experiments. The method stated "Mice were allocated randomly to experiments and groups independent of sex". However, it's unclear whether Sex is a variable for the skin phenotype.
4. Have the bulk RNA-seq and scRNA-seq data been deposited to public database? Accession numbers are not included in the Methods.

Referee #2 (Comments on Novelty/Model System for Author):

Overall, this is a major study that utilizes novel mouse models to examine scarring alopecia and its treatment. The data are

novel, extensive and mostly convincing.

Referee #2 (Remarks for Author):

This manuscript by Strobl and colleagues finds that deleting EGFR specifically in the hair follicle leads to a scarring alopecia-like phenotype in mice. The authors report that mutant hair follicle stem cells (HFSCs) have heightened JAK-STAT signaling, and that co-deleting STAT1 or JAK1/2, or depleting CD8+ T cells or NK cells can rescue hair growth. Critically, treating EGFR mutant mice with a JAK1/2 inhibitor (ruxolitinib) can also reverse the alopecia phenotype. While the data in human patients is somewhat preliminary, the findings in mice are nonetheless novel and important, as this suggests that JAK inhibition may provide a therapeutic strategy for treating scarring alopecia, which was thought to be irreversible. Overall, this is a massive, well-performed study with an immense number of experiments. The findings will be of great interest not just to dermatologists, but also to immunologists, as the authors argue that deletion of EGFR causes loss of immune privilege in the hair follicle. Although this final point needs some additional clarifications (see below), this ambitious study is highly deserving of publication in EMBO Molecular Medicine, after addressing the following points:

Major Comments

- The authors argue that alopecia is due to immune-mediated destruction of HFSCs, assessed mainly through loss of CD34+ cells by FACS. However, this may be due to downregulation of CD34, as is seen during anagen, rather than actual loss of HFSCs. Indeed, whole mounts in Fig. 1D appear to show mostly normal hair follicles in mutants, and the *Egr2-Cre;EGFR-KO* mice have prolonged anagen. Can the authors assess telogen skin in mutants prior to hair loss, and show by IHC staining that the bulge structure is actually lost? Preferably, this would involve co-staining for both CD34 and Sox9. Either positive or negative data would be helpful.
- The author perform several interesting experiments to show that different manipulations can affect MHC levels. While the figures indicate that changes in MHC were seen in HFSCs, the figure legends should state more clearly that the experiments specifically measured MHC in CD34+/CD45- bulge HFSCs. This lack of clarity occurs in figure legend 3C, 3G, 4I, 5J.
- The data from the 1 folliculitis decalvans patient treated with JAK inhibitor is unconvincing and too preliminary. Also, increased IFITM3 staining in FFA/LP is unconvincing and staining in hair follicles isn't shown. Given the very strong data from their mouse model, these patient results should be removed and saved for a future study.

Minor Comments

- Fig. 2F. Is vimentin being expressed by hair follicle cells? What Vim+ cell types are being quantitated? Could these be dermal cells that are closely associated with the hair follicle?
- Fig. 2H. How are hair follicles being quantitated? Are partial follicles counted? Skin whole mounts showing hair loss would be more convincing here, as interpretation is difficult from thin sections.
- Discussion: "In older mice (e.g. 10 months of age), the complete scarring of the hair follicle then prevents therapeutic intervention indicating the irreversible loss of the stem cell niche." This is an important point, and should be mentioned earlier in the results section, along with any experimental data.

Referee #3 (Comments on Novelty/Model System for Author):

The authors analyze the molecular-level effects of JAK inhibition in mice and human samples subjected to genetic and chemical EGFR inhibition. The manuscript represents a follow-up to their previous work in Science Translational Medicine. While the experiments are detailed and well-explained, the overall phenotypic outcome was somewhat expected based on therapeutic treatments for Alopecia Areata. However, I appreciate the identification of more detailed cellular and molecular mechanisms induced by the inhibition of the JAK/STAT1 pathway in combination with EGFR loss.

Referee #3 (Remarks for Author):

Previously, the authors demonstrated that the loss of immune privilege in hair follicle stem cells (HFSCs) induces scarring alopecia. Specifically, they showed that generic or chemically induced epidermal growth factor receptor (EGFR) inhibition triggers hair and skin inflammation associated with hair loss. In this manuscript, the authors address an open question regarding the downstream cellular and molecular mechanisms driving these skin phenotypes. They discovered that hypersensitivity to the JAK1/2-STAT1 signaling pathway exhausts the HFSC niche during inflammation, resulting in hair loss. This study expands the potential therapeutic application of JAK inhibitors to provide HFSC-specific protection against chronic folliculitis associated with EGFR inhibition in cancer therapy and scarring alopecia.

Below, I have outlined several minor points that the authors should address to enhance the clarity and robustness of the study, thereby providing a more comprehensive understanding of the mechanisms and potential therapeutic interventions.

Suggested Improvements:

1. Mouse Line Specificity (Ref 28 vs. Ref 24):

o The authors used Egr2-Cre to confer specificity to the hair follicles. However, Ref 28 suggests that Egr2-Cre affects both the interfollicular epithelium and hair follicles over time, while LRIG-1-Cre and LGR5-Cre mouse models used in Ref 24 show hair follicle specificity. Please could the authors justify the choice of Egr2-Cre in the text.

2. Quantification in Figure 1C:

o Please add quantitative data from multiple mice to support the qualitative data presented in Figure 1C. Also, specify the time point at which the images were taken.

3. Details in Figure 2D:

o Include the number of mouse models and hair follicles counted in Figure 2D. This comment applies to all figures where the data points are not clearly defined.

4. IF Analysis Timing in Figure 2F:

o Explain why the immunofluorescence (IF) analysis was performed on 2-month and 3-month-old EGFR Δ egr2 mice instead of at 1-month-old as used in the RNA analysis. Include the number of hair follicles measured and the number of mice to determine statistical significance.

5. Cartoon or H&E Staining:

o It would be beneficial to include a cartoon depicting the decline of hair follicles over time compared to wild-type follicles, providing an immediate visual representation of the observed phenotype addressed at the molecular level. Alternatively, include H&E staining of wild-type controls at corresponding time points. Please specify the number of mice analyzed in each group.

6. PROGENy Analysis Timing in Figure 3A:

o Clarify whether the PROGENy analysis was conducted at 1, 2, 3, 5, and 10 months old. Confirm if this dataset aligns with the analysis mentioned for 1-month-old mice.

7. MHC Specificity:

o Throughout the text, specify whether MHC I, MHC II, or both are being referenced.

8. Timing of Analysis in Figure 3F:

o Indicate the specific time point at which the analysis in Figure 3F was performed to correlate with the observed phenotype.

9. Statistical Significance in Figure 3B:

o Can be possible to determine if the differences observed in the heat map in Figure 3B between the triplicate are statistically significant?

10. Representation in Figure 4B:

o Clarify whether the dots in Figure 4B represent mice or hair follicles.

11. Details in Figure 5C:

o Include the number of mouse models and hair follicles counted in the figure legend of Figure 5C.

Point-by-point response to the referees**EMM-2024-20141**

We thank all reviewers for their excellent and very constructive assessment. We have included all their suggestions in the revised paper which helped to significantly improve and clarify the study. We have addressed their remarks as follows:

Referee #1 (Comments on Novelty/Model System for Author):

Ideally, the raw data may be made available to reviewers prior during the review process.

We apologize for not depositing the raw data before the review process. RNA-seq datasets are publicly available on Gene Expression Omnibus (GEO) database under RNA-Seq data:

Gene Expression Omnibus GSE273571

(<https://www.ncbi.nlm.nih.gov/geo/query/acc.cgi?acc=GSE273571>)

scRNA-Seq data: Gene Expression Omnibus GSE273572

(<https://www.ncbi.nlm.nih.gov/geo/query/acc.cgi?acc=GSE273572>)

Referee #1 (Remarks for Author):

The article EMM-2024-20141 describes a hair follicle specific EGFR knockout model that develops chronic scaling alopecia. Folliculitis is a side effect often seen in patients treated with EGFR inhibitor anticancer therapy. The manuscript contains a comprehensive and rigorous body of work. The results are interesting with clinical relevance and are well-presented.

We appreciate the referee's judgement that our manuscript contains a comprehensive and rigorous body of work and the results are interesting, well presented and clinically relevant.

Below are a few minor comments:

1. Is there direct evidence of microbial infection or dysbiosis in the EGFR^{depi} and EGFR^{dEGR2} skin? If yes, is there microbial species/strain information?

Indeed, the EGFR^{Δep} displays a microbial dysbiosis dominated by *Staphylococcus aureus*, which was published previously in Klufa and Bauer et al. Sci Transl Med 2019 (Fig. 3C, D and S3A, B). Similarly, our here developed EGFR^{ΔEgr2} mouse model also develops bacterial outgrowth and a *Staphylococcal* dominated dysbiosis (see data below).

(A) Skin swabs plated on blood agar plates from WT and EGFR Δ Egr2 littermates housed in separate cages after weaning
(B) Differential abundance plots; 16SrRNA microbiome sequencing from skin swabs of cohoused WT and EGFR Δ Egr2 littermates

Notably the Staphylococcal dominance reflects the situation in the human scarring Alopecia type: Folliculitis decalvans. If the reviewer agrees, we would like to refrain from showing these data, as we are currently working on a study detailing the anti-bacterial mechanisms downstream of EGFR, which would benefit from this description.

2. Double knockouts of EGFR and JAK2/STAT1 verified a HFSC-intrinsic JAK2/STAT1 signaling dysregulation as a key driver of folliculitis. In the animal models, JAK2/STAT1 inhibitions in young animals prevented development of folliculitis and restored hair growth in the EGFR-KO skin in 1-2M old mice. Did JAK inhibitor treatment improve hair regrowth in 5-6M old EGFRKO mice?

- Figure S5D: "Interestingly, JAK1/2 inhibition did not influence the $\square\square$ T cell compartment, the neutrophil recruitment and the NK cells (Supplemental Figure 5D)." This experiment was done with 5-6M old mice, but there was no mention about the effects of Ruxolitinib on HFSC in these old mice.

We indeed treated young mice (1-2 month old), which refers to the prophylactic setting (in order to prevent hair loss) and older mice (5-6 month old), which refers to the therapeutic setting (to re-establish hair growth). In Figure 5B – H we show 5M old EGFR Δ Egr2 mice treated for 1 month with the JAKi. Figure 5I and J shows 5 month old EGFR Δ ep mice treated for 1 month with the JAKi. In both cases hair regrowth was evident, indicating that while these mice were bald, the Sox9 positive hair follicle stem cell niche was still capable of recovery. In 10M old mice, however, were we could observe full scarring of the hair follicles, we could not re-initiate hair growth (Figure EV4E and F). Clinically our data therefore support the importance of an early treatment in human scarring alopecia before a full degradation of the

stem cell niche. We now more clearly indicate the age and treatment regime of the mice in the Figures and Figure legends. Specifically, “EGFR^{ΔEgr2} 5-month-old (5M)”, “Ctrl post-treatment day 28” and “Ruxolitinib post-treatment day 28” in Figure 5B and D, “WT 5M Ctrl, EGFR^{ΔEgr2} 5M Ctrl, WT 5M Ruxo, EGFR^{ΔEgr2} 5M Ruxo” in Figure 5C-H, “EGFR^{Δep} 5M” and WT 5M” in Figure 5I-J and “EGFR^{Δep} 5M, EGFR STAT1^{Δep} 5M, WT 5M, STAT1^{Δep} 5M” in Figure 5K-M.

- Figure S5C. P-value information is missing.

We apologize for the overlook and added the missing p-values in the Figure (now EV4C).

- Hair regrowth was not apparent in the clinical case of folliculitis decalvans 4-months after JAK inhibitor therapy. How much is known about the etiology for this patient? Does one expect to see hair growth in this patient after a longer period (e.g. 1 year) of treatment?

- JAK inhibitors are already used for various conditions of alopecia. Does the result of this study imply the importance of early treatment in scarring alopecia?

This is indeed a clinically highly relevant question. This particular patient was under various anti-microbial and anti-inflammatory treatments for over three years before successful JAKi therapy. Based on our mouse data, explained above, we hypothesize that an earlier treatment, instead of a longer treatment, would have led to hair regrowth in this patient. It is necessary to start treatment before the hair follicle stem cell niche is fully scarred and loses the capability for novel hair growth. The exact time frame for therapeutic intervention might, however, vary from mice to humans and highly likely also depends on the strength and duration of the inflammatory stimulus in the different scarring alopecia types. This emphasizes the necessity for early prediction of a successful JAKi therapy through biological markers like phospho-STAT1 for example. However, more clinical studies are necessary to fully draw this conclusion.

Since we were asked to remove this one-patient case study from the publication we added more about the patient's etiology here and not in the text:

In the parieto-occipital scalp area, there was a shiny scar plate measuring approximately 5x3 cm. At the border, multiple tufts of hair with perifollicular erythema and yellowish, firmly adherent scaling were found. There were no relevant pre-existing conditions, and the family history was negative for inflammatory skin diseases. Laboratory investigations before starting Baricitinib revealed no abnormalities. In our patient, a significant response was observed after 4 weeks of using a JAK inhibitor, with good tolerability over the 4 month treatment period.

Histopathological findings:

Basket-weave-like stratum corneum. Slightly flattened epithelium. In the dermis, the hair follicles appear rarefied. In some areas, exposed hair shafts surrounded by an inflammatory infiltrate of lymphocytes, histiocytes, and plasma cells can be seen. Fibrosis. No evidence of fungal elements in PAS staining.

3. The animal number was provided for bulk RNA-seq (n=3, WT and mutant) and scRNA-seq for 3-month-old EGFR Δ Egr2 (n=4). Is WT skin included for scRNA? The animal number information is not clear for other experiments. The method stated "Mice were allocated randomly to experiments and groups independent of sex". However, it's unclear whether Sex is a variable for the skin phenotype.

For the bulk RNAseq experiment in Figure 2 and 3, we used 3 WT and 3 mutant mice. The single cell RNAseq experiment was conducted to describe the infiltrating immune compartment in the EGFR Δ Egr2 mice. The FACS data from Figure 4A and B indicated the lack of infiltrating immune cells in WT epidermis (except for the resident LCs and DETCs, which are strongly reduced in the EGFR Δ Egr2 mice). Therefore, the analysis was performed on pooled RNA from 4 different EGFR Δ Egr2 mice to identify the transcriptional profile of their infiltrated immune compartment with focus on cytokine and chemokine expression (Figure 4D).

As we observed no sex differences in this inflammatory skin phenotype, we randomly and equally used both female and male mice for all experiments.

4. Have the bulk RNA-seq and scRNA-seq data been deposited to public database? Accession numbers are not included in the Methods.

Accession numbers are now included in the results and methods section.

RNA-seq datasets are publicly available on Gene Expression Omnibus (GEO) database under RNA-Seq data: Gene Expression Omnibus

GSE273571(<https://www.ncbi.nlm.nih.gov/geo/query/acc.cgi?acc=GSE273571>)

scRNA-Seq data: Gene Expression Omnibus

GSE273572(<https://www.ncbi.nlm.nih.gov/geo/query/acc.cgi?acc=GSE273572>)

Referee #2 (Comments on Novelty/Model System for Author):

Overall, this is a major study that utilizes novel mouse models to examine scarring alopecia and its treatment. The data are novel, extensive and mostly convincing.

We appreciate that the referee judges this study as major and values our novel mouse model of scarring alopecia.

Referee #2 (Remarks for Author):

This manuscript by Strobl and colleagues finds that deleting EGFR specifically in the hair follicle leads to a scarring alopecia-like phenotype in mice. The authors report that mutant hair follicle stem cells (HFSCs) have heightened JAK-STAT signaling, and that co-deleting STAT1 or JAK1/2, or depleting CD8+ T cells or NK cells can rescue hair growth. Critically, treating EGFR mutant mice with a JAK1/2 inhibitor (ruxolitinib) can also reverse the alopecia phenotype. While the data in human patients is somewhat preliminary, the findings in mice are nonetheless novel and important, as this suggests that JAK inhibition may provide a therapeutic strategy for treating scarring alopecia, which was thought to be irreversible. Overall, this is a massive, well-performed study with an immense number of experiments. The findings will be of great interest not just to dermatologists, but also to immunologists, as the authors argue that deletion of EGFR causes loss of immune privilege in the hair follicle. Although this final point needs some additional clarifications (see below), this ambitious study is highly deserving of publication in EMBO Molecular Medicine, after addressing the following points:

We thank the referee for pointing out that this study is massive and well-performed with great interest for dermatologist and immunologists and that it is highly suitable for EMBO Molecular Medicine.

Major Comments

- The authors argue that alopecia is due to immune-mediated destruction of HFSCs, assessed mainly through loss of CD34+ cells by FACS. However, this may be due to downregulation of CD34, as is seen during anagen, rather than actual loss of HFSCs. Indeed, whole mounts in Fig. 1D appear to show mostly normal hair follicles in mutants, and the Egr2-Cre;EGFR-KO mice have prolonged anagen. Can the authors assess telogen skin in mutants prior to hair loss, and show by IHC staining that the bulge structure is actually lost? Preferably, this would involve co-staining for both CD34 and Sox9. Either positive or negative data would be helpful.

The referee is right to point out that CD34 surface expression as assessed by FACS might vary through the hair cycle and that more stem cell markers should be included in the study. We observed that the stem cell niche develops normally in the first month of life and is slowly degraded through the second month of life during prolonged anagen, which precedes visible hair shaft loss. We now include skin sections stained for CD34 and Sox9 with quantification at various time points (2M, 3M & 5M) to describe this temporal decline of the hair follicle stem cell niche and its correlation with visible hair loss (Figure EV1G). We have also added whole mount stainings of tail epidermal sheets for Krt15 at a later time point (7M) to demonstrate complete hair follicle destruction (Figure 1I and EV1H). We observed the same phenotype with stainings for Sox9 (see below):

If the reviewer agrees, we would like to refrain from showing these latter parallel stainings with Sox9 due to space restrictions.

Whole mounts of Figure 1D depict an early timepoint 2 weeks after birth (p14), which is now indicated in Figure 1D. At this timepoint hair follicles are still intact, but surrounded by activated Langerhans cells as indicated by MHC II^{high}, Langerin expression.

- The author perform several interesting experiments to show that different manipulations can affect MHC levels. While the figures indicate that changes in MHC were seen in HFSCs, the figure legends should state more clearly that the experiments specifically measured MHC in CD34+/CD45- bulge HFSCs. This lack of clarity occurs in figure legend 3C, 3G, 4I, 5J.

We apologize for the lack of clarity and now include this detailed information in the appropriate Figure legends, specifically 3C, 3G, 4I and 5J. Briefly, MHC levels are also up-regulated in the overall epidermal compartment (as shown in Figure 3C for the EGFR Δ Egr2 model and Figure 5J for the EGFR Δ ep model). All other analysis (Figure 3G and 4I) are based on the CD34+ Scal- CD45- stem cell compartment indicating the relative loss of immune privilege.

- The data from the 1 folliculitis decalvans patient treated with JAK inhibitor is unconvincing and too preliminary. Also, increased IFITM3 staining in FFA/LP is unconvincing and staining

in hair follicles isn't shown. Given the very strong data from their mouse model, these patient results should be removed and saved for a future study.

As already explained to referee #1, due to the already far progressed disease (other therapies failed for 3 years), we could not achieve full hair regrowth. Yet, the strong inflammation as indicated by redness and yellow crust formation was effectively managed. However, we agree with the reviewer that the case report and the usage of IFITM3 as disease marker is too preliminary and removed these data from the current manuscript.

Minor Comments

- Fig. 2F. Is vimentin being expressed by hair follicle cells? What Vim+ cell types are being quantitated? Could these be dermal cells that are closely associated with the hair follicle?

We only counted Vimentin positive hair follicle cells, as indicated by Vimentin E-cadherin positivity in the hair follicles. We now indicate this more clearly in the Figure legend. However, we could also observe closely associated Vimentin positive dermal cells. As they are difficult to quantify, we now point these cells out by asterisk in the Figure and describe our finding in the Figure legend and results section of the manuscript. This phenomenon is indicative of scarring around the hair follicle (Imanishi *et al.* 2018).

- Fig. 2H. How are hair follicles being quantitated? Are partial follicles counted? Skin whole mounts showing hair loss would be more convincing here, as interpretation is difficult from thin sections.

We agree with the referee that it is indeed difficult to quantify the scarred hair follicles in 10 month old mice. We did the quantification to illustrate the degraded status of the skin and only counted clearly visible hair follicles, now indicated in the Figure EV2F by arrows. We thank the referee's suggestion to use skin whole mounts in a parallel approach. Therefore, we now also included epidermal tail whole mounts with Keratin 15 staining to show the decline of the hair follicle structure in Figure 11.

- Discussion: "In older mice (e.g. 10 months of age), the complete scarring of the hair follicle then prevents therapeutic intervention indicating the irreversible loss of the stem cell niche." This is an important point, and should be mentioned earlier in the results section, along with any experimental data.

We agree with the referee that this point is important and will underline the scarring alopecia phenotype of the mice and the necessity for early therapeutic intervention to prevent the disease to become irreversible. We now include these phenotypic data in Figure EV4E-F, which also support our finding from Figure 2H that the hair follicles are almost completely degraded in 10 month old mice (as explained in the previous point).

Referee #3 (Comments on Novelty/Model System for Author):

The authors analyze the molecular-level effects of JAK inhibition in mice and human samples subjected to genetic and chemical EGFR inhibition. The manuscript represents a follow-up to their previous work in Science Translational Medicine. While the experiments are detailed and well-explained, the overall phenotypic outcome was somewhat expected based on therapeutic treatments for Alopecia Areata. However, I appreciate the identification of more detailed cellular and molecular mechanisms induced by the inhibition of the JAK/STAT1 pathway in combination with EGFR loss.

We appreciate that the referee finds our experiments detailed and well-explained and understand that the referee feels that the therapeutic outcome is somewhat expected due to its effectiveness in non-scarring Alopecia areata. However, we want to point out that Alopecia areata is a T-cell mediated autoimmune disease and the working mechanism is hypothesized to revolve around inactivating these autoreactive T-cells. Our study describes that JAK inhibitors directly act on the hair follicles in scarring Alopecia in parallel to T-cells and NK cells. We therefore appreciate the referee's conclusion that we present a more detailed cellular and molecular working mechanism of JAK inhibition and its effect in the context of EGFR loss-of-function.

Referee #3 (Remarks for Author):

Previously, the authors demonstrated that the loss of immune privilege in hair follicle stem cells (HFSCs) induces scarring alopecia. Specifically, they showed that generic or chemically induced epidermal growth factor receptor (EGFR) inhibition triggers hair and skin inflammation associated with hair loss. In this manuscript, the authors address an open question regarding the downstream cellular and molecular mechanisms driving these skin phenotypes. They discovered that hypersensitivity to the JAK1/2-STAT1 signaling pathway exhausts the HFSC niche during inflammation, resulting in hair loss. This study expands the potential therapeutic application of JAK inhibitors to provide HFSC-specific protection against chronic folliculitis associated with EGFR inhibition in cancer therapy and scarring alopecia.

We agree with the referee that the study expands the potential therapeutic application of JAK inhibitors to protect from chronic folliculitis and scarring alopecia.

Below, I have outlined several minor points that the authors should address to enhance the clarity and robustness of the study, thereby providing a more comprehensive understanding of the mechanisms and potential therapeutic interventions.

Suggested Improvements:

1. Mouse Line Specificity (Ref 28 vs. Ref 24):

o The authors used Egr2-Cre to confer specificity to the hair follicles. However, Ref 28 suggests that Egr2-Cre affects both the interfollicular epithelium and hair follicles over time, while LRIG-1-Cre and LGR5-Cre mouse models used in Ref 24 show hair follicle specificity. Please could the authors justify the choice of Egr2-Cre in the text.

2. Quantification in Figure 1C:

o Please add quantitative data from multiple mice to support the qualitative data presented in Figure 1C. Also, specify the time point at which the images were taken.

The referee is right that the Egr2-cre activity, although very specific to the upper hair follicle during the first month of life, spills over to the epidermis as published in Ref 28. However, we did not observe robust EGFR deletion in the epidermal compartment as opposed to the upper hair follicle even at 3 months of age. We now clearly specify the age in Figure 1C. As the referee suggested in 2., we now add quantitative data to Figure EV1D to support our observation.

Lrig1- and Lgr5-cre lines are indeed more specific to the respective hair follicle compartments. However, as these cre-lines are tamoxifen inducible and we need a very robust deletion during hair shaft eruption around P5 to generate the folliculitis driving the scarring alopecia, it is necessary to perform tamoxifen injections starting from birth. This procedure unfortunately introduces side effects that drastically impact on the survival of these young mice, rendering these model systems not ideal for more complex experiments. However, we do see the same hallmarks (i.e. MHC I up-regulation and CD34 decline) as in the more robust and reproducible Egr2-cre model (see below).

Epidermal cells by FACS of older than 5M EGFR^{ALRIG1/LGR5} mice and littermates
(A) % of CD45+ immune cells
(B) Geometric Mean of CD34 of CD45- keratinocytes, and % MHC-I of CD34+ HFSC

If the reviewer agrees, we would like to refrain from showing these data due to space restrictions.

3. Details in Figure 2D:

o Include the number of mouse models and hair follicles counted in Figure 2D. This comment applies to all figures where the data points are not clearly defined.

We apologize for not specifying these details properly and now add the information to the Figure legends (Figure 2D, 2F and 5D). Specifically, every dot represents mean data from one independent mouse. Cells from more than three hair follicles were counted per mouse.

4. IF Analysis Timing in Figure 2F:

o Explain why the immunofluorescence (IF) analysis was performed on 2-month and 3-month-old EGFR Δ egr2 mice instead of at 1-month-old as used in the RNA analysis. Include the number of hair follicles measured and the number of mice to determine statistical significance.

The RNA seq analysis from sorted CD34+ Scal- CD45- hair follicle stem cells was performed to reveal the overall transcriptional landscape of the stem cell niche before its decline. As can be seen in Figure 2D (compare data from 1 and 2M), we observed that the transcriptional profile translates more robustly to the protein and cell-effect level (proliferation, apoptosis and fibrosis) at later time-points. However, we want to stress that these later time-points are still within the active disease progression phase indicated by a detectable but impaired stem cell niche. We now better explain this in the respective results part of Figure 2.

5. Cartoon or H&E Staining:

o It would be beneficial to include a cartoon depicting the decline of hair follicles over time compared to wild-type follicles, providing an immediate visual representation of the observed phenotype addressed at the molecular level. Alternatively, include H&E staining of wild-type controls at corresponding time points. Please specify the number of mice analyzed in each group.

We thank the referee for this productive suggestion, which provides a better overview of the temporal decline during scarring hair follicle destruction and now added a illustration in Figure 1H and H&E stainings from the individual time-points to Figure EV1G.

6. PROGENy Analysis Timing in Figure 3A:

o Clarify whether the PROGENy analysis was conducted at 1, 2, 3, 5, and 10 months old. Confirm if this dataset aligns with the analysis mentioned for 1-month-old mice.

The PROGENy analysis was indeed carried out with the same dataset produced for Figure 2 (1 month old mice) and indicates the early dysregulation of the JAK pathway before the decline of the stem cell niche. We now clarified this in the results part of the manuscript and the respective Figure legend 3A.

7. MHC Specificity:

o Throughout the text, specify whether MHC I, MHC II, or both are being referenced.

We apologize for this overlook on our side and now specified it throughout the manuscript.

8. Timing of Analysis in Figure 3F:

o Indicate the specific time point at which the analysis in Figure 3F was performed to correlate with the observed phenotype.

These pictures were taken from 5 month old mice. We now include this information directly in the Figure for clarity.

9. Statistical Significance in Figure 3B:

o Can be possible to determine if the differences observed in the heat map in Figure 3B between the triplicate are statistically significant?

We now indicate the statistical significance next to the heatmap for each gene in Figure 3B.

10. Representation in Figure 4B:

o Clarify whether the dots in Figure 4B represent mice or hair follicles.

11. Details in Figure 5C:

o Include the number of mouse models and hair follicles counted in the figure legend of Figure 5C.

Ad Figure 4B and 5C: Each dot represents FACS data from one individual mouse. For the IF quantifications in 5D, cells of more than three hair follicles were counted per individual mouse. Each dot represents mean value from one mouse. We now clarified this in the respective Figure legends.

9th Oct 2024

Dear Dr. Bauer,

Thank you for submitting your revised study, and please accept my apologies for the delay in getting back to you as I was traveling for work. We have now received the reports from the three referees who evaluated your revised manuscript. As you will see from the reports below, they are satisfied with the revisions, and I will therefore be able to accept your manuscript once the following editorial issues will be addressed:

1/ Manuscript text:

- Please remove the highlights in the text and only keep in track changes mode any new modification.
- The following emails bounced, please correct: Regina Jin (regina.jin@medunwien.ac.at) and Jörg Klufa (joerg.klufa@boehringer-ingenelheim.com).
- Please correct the order of the manuscript sections as follows: Abstract / Keywords / Introduction / Results / Discussion / Materials and Methods / Acknowledgements / Disclosure and competing interests statement / The Paper Explained / For More Information / References / Figure legends / Tables and their legends / Expanded View Figure legends.
- Methods:
 - o Thank you for providing a reagents and tools table, please remove it from the manuscript and upload it as a separate file.
 - o Patient samples: please include a statement that the experiments conformed to the principles set out in the WMA Declaration of Helsinki and the Department of Health and Human Services Belmont Report.
 - o Antibodies: please provide dilutions/concentrations.
 - o Cells: please indicate whether the cells were tested for mycoplasma contamination (and correct the checklist accordingly).
 - o Please rename "Data and Materials Availability" to "Data Availability" and place it after the Methods section. Please remove "All other data are available in the main text or the supplementary materials."
- Please merge the Acknowledgements and Funding sections.
- Author contributions: CRediT has replaced the traditional author contributions section because it offers a systematic machine readable author contributions format that allows for more effective research assessment. Please remove the Authors Contributions from the manuscript and use the free text boxes beneath each contributing author's name in our system to add specific details on the author's contribution. More information is available in our guide to authors.
- Data citation: the (Data ref: Cohen et al., 2024) data citation is not tagged with the label "DATASET" in the reference list, please correct. Please note that a generic URL is missing for (Data ref: Cohen et al., 2024) data citation.

2/ Figures:

- There is a callout for a Supplementary Fig 4, please correct.
- Figure re-use should be indicated in the legends (i.e. Figure 6C H&E and Figure EV5 H&E).
- Please address the queries from our copy editors in the figure legends:
 1. Please note that the exact p values are not provided in the legends of figures 1b, f-g, i; 2d, f, h; 3b, d-e, g-h; 4b, e, h-j; 5a, c-j, l-m; EV 1b, d, f; EV 2a, i-j; EV 3d; EV 4a-d, g-h.
 2. Please indicate the statistical test used for data analysis in the legends of figures EV 2b, d; EV 3a.
 3. Please note that in figures 5a, c, e, h, m; EV 2i-j; EV 3d; EV 4b-d, h; there is a mismatch between the annotated p values in the figure legend and the annotated p values in the figure file that should be corrected.
 - 4 Please note that information related to n is missing in the legend of figure EV 5c.
 5. Please note that scale bar and its definition are missing for figures 6a; EV 5b.
 6. Please note that the white/yellow dotted line is not defined in the legend of figure 6a, EV 1g. This needs to be rectified.
 7. Please note that the white/black arrowheads are not defined in the legend of figure 1c; 2d, f; EV 2f. This needs to be rectified.

3/ Source Data:

Please upload them as 1 file per figure, with numerical data included for each figure. Files that are too large to upload should be deposited externally and a link should be added in the Data Availability section.

4/ Synopsis:

I introduced minor modifications in your text to fit our format, please let me know if you agree or amend as you see fit: "Defective EGFR signalling induces hypersensitive JAK-STAT1 pathway in the hair follicle. This results in the collapse of the stem cell niche during microbial skin inflammation and the scarring of the hair follicle. JAK inhibition therefore offers a therapeutic option to counteract scarring alopecia.

- Chronic microbial folliculitis associated with EGFR deficiency initiates hallmarks of scarring alopecia
- Cell intrinsic JAK1/2-STAT1 signalling drives hair follicle destruction
- CD8 T-cells and NK-cells expressing IFN trigger the JAK-STAT1 cascade
- Therapeutic JAK1/2 inhibition alleviates skin inflammation, restores barrier integrity and leads to hair regrowth"

Thank you for providing a nice synopsis picture. Please upload it as an independent png/jpeg/tiff file px wide x 300-600 px high. A cropped portion of this image will serve as thumbnail for the table of content on our webpage.

5/ As part of the EMBO Publications transparent editorial process initiative (see our Editorial at <http://embomolmed.embopress.org/content/2/9/329>), EMBO Molecular Medicine will publish online a Review Process File (RPF) to accompany accepted manuscripts.

This file will be published in conjunction with your paper and will include the anonymous referee reports, your point-by-point response and all pertinent correspondence relating to the manuscript. Let us know whether you agree with the publication of the RPF and as here, if you want to remove or not any figures from it prior to publication.

I look forward to receiving your revised manuscript.

Yours sincerely,

Lise Roth

***** Reviewer's comments *****

Referee #1 (Comments on Novelty/Model System for Author):

My comments are fully addressed.

Referee #2 (Remarks for Author):

This revised manuscript by Strobl et al., addresses the previous critiques. Overall, this is a thorough and extensive study, and provides excellent mechanistic insight into an EGFR mutant mouse model of scarring alopecia. I fully support publication in EMBO Molecular Medicine.

Referee #3 (Comments on Novelty/Model System for Author):

The mouse model used in this study is well-suited for investigating chronic scarring alopecia. It closely mimics the human disease pathology, including the involvement of key signaling pathways like JAK/STAT. This makes it highly relevant for understanding the cellular and molecular mechanisms underlying this disease. Additionally, the mouse model allows for precise chemical and genetic manipulation, enabling the authors to perform in vivo knockouts to identify potential therapeutic targets.

Referee #3 (Remarks for Author):

I have carefully evaluated the authors' responses to the concerns raised in the first review, and I believe that they have addressed most of the issues adequately.

This study provides novel and valuable insights into the cellular and molecular mechanisms of chronic scarring alopecia. The authors successfully demonstrate that the JAK/STAT1 signaling pathway is a sensitive target for scarring alopecia, which aligns with the scope of EMBO Molecular Medicine in terms of uncovering molecular mechanisms with potential clinical relevance. The translational potential of their findings further strengthens the significance of this manuscript, making it a strong candidate for the journal.

Overall, I am satisfied with the revisions and believe the manuscript is now suitable for publication.

The authors addressed the minor editorial issues.

24th Oct 2024

Dear Dr. Bauer,

Thank you for submitting your revised files. I am pleased to inform you that your manuscript is accepted for publication and is now being sent to our publisher to be included in the next available issue of EMBO Molecular Medicine!

I have cropped a small portion of your synopsis (attached) to be displayed as a thumbnail on our website. Please let me know immediately if you don't approve, as changes at proofing are usually not allowed.

We note that you suggested a nice cover image, and we'll get back to you once the issue is assembled.

If you have any questions, please do not hesitate to contact the Editorial Office.

Thank you for your nice contribution to EMBO Molecular Medicine.

Yours sincerely,

Lise
